



# Analysis of Flash Drought in China using Artificial Intelligence models

Linqi Zhang[1, 2], Yi Liu[1, 2*], Liliang Ren[1, 2*], Adriaan J. Teuling[3], Ye Zhu[4], Linyong Wei[1], Linyan Zhang[1], Shanhu Jiang[1], Xiaoli Yang[1], Xiuqin Fang[1], Hang Yin[5]

[1]State Key Laboratory of Hydrology-Water Resources and Hydraulic Engineering, Hohai University, Nanjing 210098, China
[2]College of Hydrology and Water Resources, Hohai University, Nanjing 210098, China
[3]Hydrology and Quantitative Water Management Group, Wageningen University, Wageningen 6708PB, The Netherlands
[4]College of Hydrology and Water Resources, Nanjing University of Information Science & Technology, Nanjing 210044, China
[5]Institute of Water Resources for Pastoral Area, Ministry of Water Resources, Inner Mongolia 010020, China

*Correspondence to*: Yi Liu (liuyihhdx@126.com) & Liliang Ren (RLL@hhu.edu.cn)

**Abstract.** The term "Flash drought" describes a type of drought with rapid onset and strong intensity, which is co-affected by both water-limited and energy-limited conditions. It has aroused widespread attention in related research communities

due to its devastating impacts on agricultural production and natural system. Based on a global reanalysis dataset, we identify flash droughts across China during 1979~2016 by focusing on the depletion rate of weekly soil moisture percentile. The relationship between the rate of intensification (RI) and nine related climate variables is constructed using three artificial intelligence (AI) technologies, namely, multiple linear regression (MLR), long short-term memory (LSTM), and random forest (RF) models. On this basis, the capabilities of these algorithms for estimating RI and droughts (flash droughts and

traditional slowly-evolving droughts) detection were analyzed. Results showed that the RF model achieved the highest skill in terms of RI estimation and flash droughts identification among the three approaches. Spatially, the RF-based RI performed best in the southeastern China, with an average CC of 0.90 and average RMSE of 2.6th percentile per week, while the poor performances were found in Xinjiang region. For drought detection, all three AI technologies presented a better performance in monitoring flash droughts than in conventional slowly-evolving droughts. Particularly, the probability of

detection (POD), false alarm ratio (FAR), and critical success index (CSI) of flash drought derived from RF were 0.93, 0.15, and 0.80, respectively, indicating that RF technology is preferable to estimate the RI and monitoring flash droughts by considering multiple meteorological variable anomalies in adjacent weeks of drought onset. In terms of the meteorological driving mechanism of flash drought, the negative precipitation (P) anomalies and positive potential evapotranspiration (PET) anomalies exhibited a stronger synergistic effect on flash droughts comparing to slowly-developing droughts, along with

asymmetrical compound influences in different regions over China. For the Xinjiang region, P deficit played a dominant role in triggering the onset of flash droughts, while in the southwestern China, the lack of precipitation and enhanced evaporative demand almost contributed equally to the occurrence of flash drought. This study is valuable to enhance the understanding of flash drought and highlight the potential of AI technologies in flash droughts monitoring.





# 1 Introduction

Drought is generally regarded as a slowly-evolving climate phenomenon, which may persist for several months or even years (Allen et al., 2010; Mishra and Singh, 2010). Several recent studies suggested that drought can also develop in a more intense and quicker manner under extreme atmospheric anomalies (Ford and Labosier, 2017; Hunt et al., 2014; Otkin et al., 2013). For instance, large precipitation deficits or increase in evaporative demand derive from unusual climate conditions (e.g., enhanced air temperatures, strong wind, or low humidity). This type of drought is usually termed as "flash drought",

which has been used to describe an additional type of drought with the characteristic of rapid onset and high intensification (Senay et al., 2008; Svoboda et al., 2002). Comparing to the conventional droughts, flash droughts may lead to severer impacts on agricultural production and natural systems due to their sudden-onset nature which makes it difficult to provide early warning and effective countermeasures for governors and stakeholders (Anderson et al., 2013). For example, the summer drought in 2012 that occurred across the central United States was recognized as a historic flash drought event,

which led to considerable damage to local crops with $12 billion economic losses (Hoerling et al., 2014). Therefore, it is an urgent need to improve the understanding of flash droughts, take effective measures to identify them, and conduct the simulation analysis of flash droughts.

Flash drought, as an active topic of drought research, has aroused increasing attention by the scientific community over

recent years. However, there is no consistent standard on how we recognize and define flash droughts. One representation is proposed by Mo and Lettenmaier (2015, 2016), which combines several thresholds for hydrometeorological variables including soil moisture, precipitation, temperature, and evapotranspiration. Based on their method, two types of flash droughts can be distinguished: the precipitation deficit flash drought (PDFD) and the heat wave flash drought (HWFD). The former type was triggered by negative precipitation anomalies, while the latter type was driven by high temperature or/and

heat wave. In a different manner, Ford and Labosier (2017) suggested the rapid decline rate of soil moisture is an important feature that distinguished from traditional slowly-evolving droughts, and defined a flash drought event as soil moisture to reduce from above the 40th percentile to below the 20th percentile within 4 pentads. Liu et al. (2020a) identified flash droughts from the perspective of rapid intensification of soil moisture and compared the results with those from the PDFD and HWFD identification approaches over the Yellow River basin. Oktin et al. (2018) stated that the approach of flash

drought identification should account for two aspects, one refers to the rapid intensification that can reveal the characteristic of 'flash', and the other is the actual moisture limitation condition (i.e., drought severity), which can reflect the feature of 'drought'. Besides, several researches applied drought indices to recognize flash drought events, such as Evaporative Stress Index (ESI, Anderson et al. 2013), Standardized Evaporative Stress Ratio (SERS, Christian et al., 2019), Standardized Precipitation Evaporation Index (SPEI, Noguera et al., 2020), Evaporative Demand Drought index (EDDI, Pendergrass et al.,

2020), and Soil Moisture Volatility Index (SMVI, Osman et al., 2021). Among these literatures, soil moisture was a commonly used variable for flash drought identification due to its important role for controlling the exchange of water and



heat in the process of land-atmosphere feedbacks (AghaKouchak et al., 2015; Ford et al., 2015; Hunt et al., 2009; Yuan et al., 2017).

The fifth assessment report (AR5) of the Intergovernmental Panel on Climate Change (IPCC) provided a comprehensive assessment for recent and future changes in various types of droughts, and suggested that they should be considered separately (IPCC, 2013). Climate change has risen the temperature of land surface, which has led droughts to occur in a manner of higher frequency and greater intensity (Trenberth et al., 2014). Moreover, in the context of global warming, high temperature and heat wave occur more frequently due to land-atmosphere interaction, providing a favourable environment
for the rapid intensification of drought (Teuling et al., 2018; Wang et al., 2016). From the perspective of physical mechanisms, the evolution of flash drought involves complicated processes. Though a lack of precipitation for a certain period is a necessary requirement for droughts to develop, precipitation deficit alone is not likely to induce flash droughts (Otkin et al., 2018). Rather, the joint efforts of multiple meteorological variables, e.g., a lack of precipitation, enhanced evaporative demand caused by unusual high temperature, low humidity, strong wind, and sunshine duration, are possibly to
induce a rapid intensification in soil moisture (Hobbins et al., 2016). In other words, the occurrence of flash droughts is related to a variety of climate variables associated with water-limited and energy-limited conditions (Pendergrass et al., 2020).

In the context of global climate change, China has also experienced flash droughts frequently in recent years (Feng et al.,
2014; Sun and Yang, 2012; Wang et al., 2011; Yuan et al., 2015). For example, the 2013 summer drought influenced 13 provinces in the southern China and caused a great loss for Guizhou and Hunan province with the damage of over 2 million hectares of crops. To improve the understanding of short-term droughts across China, Wang et al. (2016) applied temperature, evapotranspiration, and soil moisture anomalies to examine the variabilities of flash droughts and reveal their increasing trends mainly related to long-term warming. Liu et al. (2020b) investigated the temporal and spatial distribution
of flash droughts over China from 1979 to 2018 and analyzed the coexisting relationship between flash droughts and seasonal droughts. It is necessary to further enrich the knowledge of flash droughts and their mechanisms for the sake of better guiding the development of early warning systems on droughts. There has been limited studies to date in regards to monitor and simulate flash droughts from a climatic perspective, especially for China with its strong climate gradients and complicated spatial heterogeneity.


Artificial intelligence (AI) technologies, as the well-known data-driven methods, provide an opportunity to describe and predict complicated physical processes based on a combination of abundant data and advanced model architectures (Kadow et al., 2020; Pan et al., 2019; Pradhan et al., 2020). In recent years, AI models had achieved considerable progresses in the hydrological process (Bennett et al., 2021; Kim et al., 2021), climate change (Li et al., 2020; Mokhtar et al., 2021), earth
system research (Cui et al., 2016; Zhang et al., 2021) and their sub-fields owing to their efficient computation and self-

learning intelligence. Among various options, three AI technologies are mostly used, i.e., the multiple linear regression (MLR), long short-term memory (LSTM), and random forest (RF). MLR is one of the simplest artificial intelligence algorithms due to its simple construction and short computation cost. LSTM is a special type of Recurrent Neural Network (RNN) with added memory structures by introducing several gates, for instance, input gate, forget gate, and output gate

(Hochreiter et al., 1997). As for the RF model, it is a nonparametric and ensemble machine learning technology in a combination of the concepts of decision trees and bagging, which was widely applied in classification, regression, and other tasks (Breiman et al., 2001; Chen et al., 2019; Hutengs and Vohland, 2016). However, few studies used them to simulate and monitor flash droughts.

The objectives of this study are fourfold: to identify flash drought across China form the perspective of rapid intensification of soil moisture, and to evaluate the performance of the MLR, LSTM, and RF models in estimating RI, and to explore their capabilities for flash droughts detection, and to explore the relationship between RI and climate drivers. The remainder of this work is organized as follows. Section 2 and Section 3 provides a brief introduction of the study area, dataset collecting and processing, and the method for identifying flash droughts. In Section 4, we present the evaluation of RI simulation

results, the performance comparison of AI technologies in terms of flash droughts and slow evolving droughts, as well as a specific investigation on typical flash drought events. Section 5 discusses the potential reasons for the varied performances of AI models in RI estimation, and their feasibilities in flash droughts detection. Finally, the main conclusions are given in Section 6.

## 2 Study area and data

### 2.1 Study area

China is located in the east of Asia and borders the western bank of the Pacific Ocean (3°51′N-53°33′N and 73°33′E-135°05′E). It has a vast spatial extent, covering an area of about 9.6 million km$^2$. From west to east, the elevation is gradually decreased and ranges from 0 to 8377 m. There are five primary terrain types in this study area, including plateau, plain, mountain, hill, and basin. According to the spatial distribution of the annual average precipitation, mountain ranges

and elevations (Chen et al., 2013), we divided China into eight subregions, i.e., Northeast China (NE), Northern China (NC), the middle and lower reaches of the Yangtze River regions (MLYR), Southeastern China (SE), Northwestern China (NW), Southwestern China (SW), Qinghai-Tibet Plateau (QTP), and Xinjiang (XJ), to analyze the spatial heterogeneity of RI.



## 2.2 Data acquisition and processing

### 2.2.1 ERA-Interim soil moisture

ERA-Interim SM reanalysis product was released from the European Center for Medium-Range Weather Forecast (ECMWF; https://apps.ecmwf.int/datasets/data/interim-full-daily/levtype = sfc/). It is produced by driving the Title ECMWF Scheme for Surface Exchange over Land (TESSEL) model with the meteorological forcing derived from ERA-Interim atmospheric reanalysis. The datasets provide daily SM data coving the period of 1979 to the present at 75 km spatial resolution. The volumetric SM was obtained at four soil depths (i.e., 0-7, 7-28, 28-100, 100-289 cm). Meanwhile, ECMWF could provide

SM at different spatial resolutions based on its platform for optional interpolation calculation. In this study, the daily SM data of the top layer (0-7 cm) at a spatial resolution of 0.25°during 1979-2016 were collected and they were generated into weekly values for intercomparison.

### 2.2.2 Meteorological forcing

Daily point-scale meteorological observations, including precipitation (P), average air temperature ($T_{mean}$), maximum

air temperature ($T_{max}$), minimum air temperature ($T_{min}$), air pressure (PRS), relative humidity (RHU), wind speed (WIN), sunshine duration (SSD), from 756 national stations were employed. All these data have complete records from 1979 and 2016 and can be acquired from the China Meteorological Administration website (CMA, http://data.cma.cn/). The potential evapotranspiration (PET) was calculated using the physically-based Penman equation (Penman, 1948) with a variety of meteorological variables such as air temperature, RHU, and WIN involved.

These point-based data were interpolated into gridded data at a spatial resolution of 0.25° by the method of inverse distance weighted (IDW).

## 3 Methodology

### 3.1 Flash drought identification

There is no consistent definition of flash drought. In this study, we adopt a quantitative method to identify flash

droughts by focusing on the rate of intensification (RI) during their onset-development phase (Liu et al., 2020a). Fig. 1 depicts the unusually rapid development process of a flash drought characterized by the significant depletion of soil moisture percentile and the anomalies of precipitation, temperature, and potential evapotranspiration in the adjacent weeks of drought onset. The upper limit (see the yellow line in Fig. 1a) represents the threshold of the 40th percentile that the soil is suffering abnormally dry conditions, while the lower limit (see the red line in Fig. 1a) denotes the 20th

percentile when moisture deficits have the potential to cause severe impacts on the environment. As shown in Fig. 1, precipitation presents negative anomalies and positive anomalies are found for $T_{max}$ and PET in the onset-development





phase, this leads to a sharp reduction for the soil moisture percentile from above 40th to 5th percentile within 3 weeks. Supposing $T_0$ is the onset time when drought occurs, and $T_n$ denotes the termination time for the onset-development stage when the rapid decline of soil moisture ceases but turn to smooth fluctuations or even an increased tendency

instead. $T_n$ can be determined through a polynomial function and located when the first derivative of the constructed polynomial equals zero in calculus (Liu et al. 2020a). With the onset time and termination time, and the intensification rate of a drought event can be calculated as:

$$RI = \frac{1}{n}\sum_{i=1}^{n}\left[\frac{SM(t_{i+1})-SM(t_i)}{t_{i+1}-t_i}\right], \quad t_1 \leq t_i \leq t_n, \tag{1}$$

$$s.t = \{\min[SM(t_i)] \leq 20th\}, \tag{2}$$

Where $t_1$ is the onset time, $t_n$ denotes the termination time, $SM(t_i)$ is the soil moisture percentile at time $t_i$ in the rapid intensification process of drought.

In this method, a flash drought event is recognized when RI exceeded a predetermined threshold. We followed the suggestion of Liu et al. (2020a) by using a criterion of -6.5th percentile per week to identify flash drought events. This value is comparable to the criterion suggested by Ford and Labosier (2017), who defined a flash drought event as soil

moisture percentile decreases from above the 40th percentile to blow the 20th percentile within 20 days. In this study, we used the absolute value of RI to indicate the depletion rate of soil moisture percentile for expression convenience, i.e., a flash drought event was recognized when RI exceeded 6.5th percentile per week. Besides, the nonlinear relationship between RI and nine meteorological variables in the adjacent weeks ($T_{0-7}$ ~ $T_{0+7}$) was constructed based on the RF models.



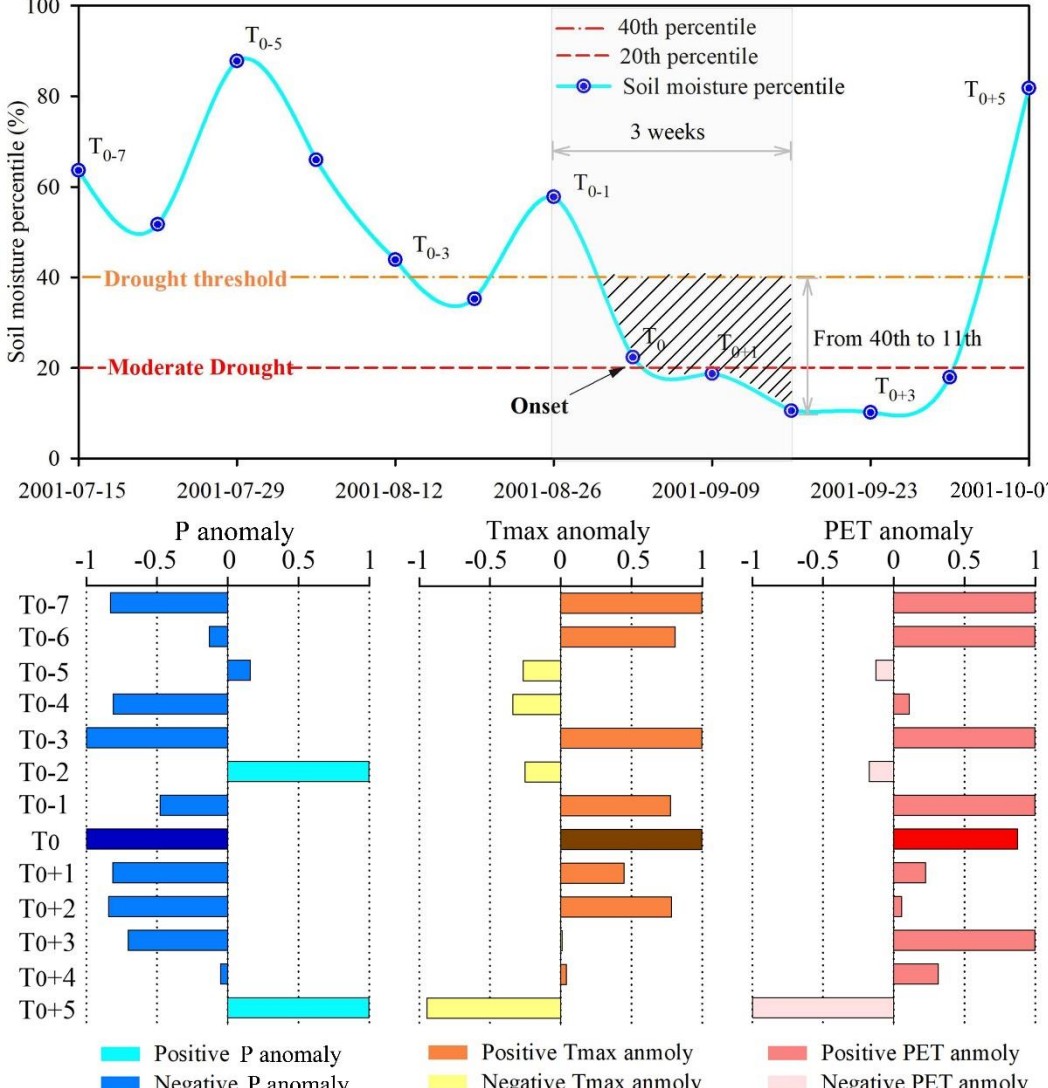


**Figure 1:** A concept map for identifying flash droughts. (a) The evolution process of flash drought identified by the rapid depletion of soil moisture percentile; $t_0$ denotes the drought onset time; $t_{0+2}$ represent the termination time where the rapid decline of soil moisture ends; The $T_{0-7}$~$T_{0-1}$ denotes 1~7 weeks prior to $T_0$, while $T_{0+1}$~$T_{0+5}$ represents the lagged 1~5 weeks of $T_0$. The period $T_0$~$T_{0+3}$ is the onset-development stage of flash drought. Data are from the grid cell (39.875°N, 116.375°E) where the Beijing city is located. (b) The bar of the anomaly values of three hydrometeorological variables (i.e., precipitation, maximum temperature, and potential evapotranspiration) in the adjacent weeks of drought onset ($T_{0-7}$~$T_{0+5}$). The light color represents positive anomaly, while the dark color denotes negative anomaly.



## 3.2 Multiple linear regression

The multiple linear regression (MLR) model is usually utilized to describe the linear relationship between the independent variables and dependent variables. Meteorological variables including P and RHU (reflecting the moisture status), and seven energy-related factors including PET, $T_{mean}$, $T_{max}$, $T_{min}$, PRS, WIN, and SSD in the adjacent weeks ($T_{0-7} \sim T_{0+7}$) of drought onset were employed as independent variables, while the observed RI was set as a dependent variable. The MLR was employed to construct the linear relationship between the observed RI and meteorological

anomalies through the following equation:

$$RI_i = \alpha_0 + \alpha_1 X_{1i} + \cdots + \alpha_j X_{ji} + \cdots + \alpha_n X_{ni} \ (i = 1, 2, \ldots, m; \ j = 1, 2, \ldots, n)$$

$$RI = \begin{bmatrix} RI_1 \\ RI_2 \\ \vdots \\ RI_m \end{bmatrix}, X = \begin{bmatrix} 1 & X_{11} & \ldots & X_{n1} \\ 1 & X_{12} & \ldots & X_{n2} \\ \vdots & \vdots & \vdots & \vdots \\ 1 & X_{1m} & \ldots & X_{nm} \end{bmatrix}, \alpha = \begin{bmatrix} \alpha_0 \\ \alpha_1 \\ \vdots \\ \alpha_n \end{bmatrix} \tag{3}$$

where $X_{ji}$ represents the anomaly value for meteorological variable $j$ in the drought event $i$; $\alpha_0$ and $\alpha_j$ are intercept and corresponding regression coefficients, respectively; $m$ is the number of drought events at a given grid cell; $n$ is the

number of input variables in the adjacent weeks of drought onset time; $RI_i$ represents the estimated RI for a drought event $i$ at a given grid cell based on the MLR method. The corresponding regression coefficients in each equation can reflect the importance of independent variable to dependent variable, which has the same function as regression weights. The importance of meteorological variables (i.e., P and PET) to RI would be presented in the Discussion section.

## 3.3 Long short term memory


   Long-short term memory (LSTM) proposed by Hochreiter et al. (1997) is a special type of Recurrent Neural Network (RNN). Compared with traditional RNNs, it has memory structures that can combine previous information into the current time step for dealing with long-term dependencies between input and output features. The input of LSTM cells is composed of three parts: input vector at the current time $x^{(t)}$, the output of LSTM cell at the previous time $h^{(t-1)}$,

and cell state at the last time $c^{(t-1)}$. LSTM cell has two output values: the output of LSTM cell at current time $h^{(t)}$ and current cell state $c^{(t)}$. Each LSTM cell has three gates: input gate $i^{(t)}$, forget gate $f^{(t)}$, and output gate $o^{(t)}$. The input gate decides what new information would be added to the current cell state $x^{(t)}$, the forget gate determines how much of the previous cell state needs to be forgotten by a sigmoid function between the input for the current time $x^{(t)}$ and the previous output $h^{(t-1)}$, and the output gate controls the retention degree of the cell state to $h^{(t)}$ in the current time. $\tilde{c}_t$ is

the candidate of new cell state values, which is calculated by a sigmoid function with a linear relationship on $x^{(t)}$ and $h^{(t-1)}$. The cell state for the current time is updated after $\tilde{c}_t$ is attained. These formulas were described as follows:





$$i^{(t)} = \sigma(w_i x^{(t)} + u_i h^{(t-1)} + b_i) \tag{4}$$

$$f^{(t)} = \sigma(w_f x^{(t)} + u_f h^{(t-1)} + b_f) \tag{5}$$

$$\tilde{c}^{(t)} = tanh(w_c x^{(t)} + u_c h^{(t-1)} + b_c) \tag{6}$$

$$c^{(t)} = f^{(t)} \otimes c^{(t-1)} + i^{(t)} \otimes \tilde{c}^{(t)} \tag{7}$$

$$o^{(t)} = \sigma(w_o x^{(t)} + u_o h^{(t-1)} + b_o) \tag{8}$$

$$h^{(t)} = o^{(t)} \otimes tanh(c^{(t)}) \tag{9}$$

Where $\sigma$ is the sigmoid function $\sigma = \frac{1}{1+e^{-x}}$, $\otimes$ is element wise multiplication, $w_s$ (i.e., $w_i, w_f, w_c, w_o$) are the matrices of the weights from the input gate $i^{(t)}$, forget gate $f^{(t)}$, cell state $c^{(t)}$, output gate $o^{(t)}$ to the input, respectively,

$\mu_s$ (i.e., $\mu_i, \mu_f, \mu_c, \mu_o$) are the weight matrices from the input gate $i^{(t)}$, forget gate $f^{(t)}$, cell state $c^{(t)}$, output gate $o^{(t)}$ to the hidden layer, respectively, $b_s$ (i.e., $b_i, b_f, b_c, b_o$) are bias parameters associated with the input gate $i^{(t)}$, forget gate $f^{(t)}$, cell state $c^{(t)}$, output gate $o^{(t)}$. $w_s, \mu_s$ and $b_s$ are adjusted using back propagation through time in the training period.

### 3.4 Random forest

Random Forest (RF) proposed by Breiman (2001) is a nonparametric and ensemble machine learning technology that combines the concepts of decision trees and bagging. It can be applied in classification, regression, and other tasks due to its important capabilities in capturing the complex nonlinear interactions between the target variable and the response variables (Hutengs and Vohland, 2016). For a regression task, the construction of the RF method consists of three steps: (1) This algorithm classified the input data into many decision trees. Each of them is made up of a root

node, internal nodes, and leaf nodes, and built from a bootstrap sample that contains a random subset of input data and a random subset of target variables. The left samples in each bootstrap sample process, so-called the out-of-bag or OOB samples, are an important feature of RF and will be not included in the model construction. The OOB can be applied to examine the performance of the constructed model, and the mean squared error (MSE) based on OOB samples can be used for testing error estimation. (2) All the decision trees make up a forest and each tree in the forest

has a predicted value. (3) The final outputs of the RF method are produced by the aggregation of the prediction value of all the individual tree. In terms of key parameters of RF regression model, the minimum sample leaf, the number of decision trees and feature are needed to set. In this study, we set the minimum sample leaf between 50-150, and the number of decision trees and the feature were set to 3 and 1000, respectively, according to stabilizing results of the OOB error. Details of RF methods and corresponding parameters are referred to Breiman (2001) and Hastie et al.

240 (2008).





## 3.5 Evaluation metrics

The objectives of this study were to evaluate the performances of MLR, LSTM, and RF in estimating the RI of drought events and assess their capabilities in capturing flash droughts. Four evaluation metrics were employed: the correlation coefficient (CC) was used to assess the consistency between the simulated and observed RI, with a perfect value of 1;

the root mean squared error (RMSE) and mean error (ME) can estimate their errors with an optimal value of 0; the relative bias (BIAS) was employed to calculate the deviations of the simulated RI from observed RI, with an excellent value of 0. These evaluation metrics were specified by equations 10-13 as below:

$$CC = \frac{\sum_{i=1}^{n}(RI_{obs}(i)-\overline{RI_{obs}})(RI_{smi}(i))-\overline{RI_{smi}}}{\sqrt{\sum_{i=1}^{n}(RI_{obs}(i)-\overline{RI_{obs}})^2}}, \quad (10)$$

$$RMSE = \sqrt{\frac{1}{n}\sum_{i=1}^{n}(RI_{smi}(i)-RI_{obs}(i))^2}, \quad (11)$$

$$ME = \frac{1}{n}\sum_{i=1}^{n}(RI_{smi}(i)-RI_{obs}(i)), \quad (12)$$

$$BIAS = \frac{\sum_{i=1}^{n}(RI_{smi}(i)-RI_{obs}(i))}{\sum_{i=1}^{n}RI_{obs}(i)}, \quad (13)$$

where $RI_{obs}(i)$ is the observed RI at grid $i$, $RI_{smi}(i)$ is the simulated RI at grid $i$, $\overline{RI_{obs}}$ is the mean observed RI value, $\overline{RI_{smi}}$ is the mean simulated RI value, and $n$ is the number of samples.

In addition, three skill scores, including the probability of detection (POD), false alarm ratio (FAR), and critical

success index (CSI), were employed to measure the performances of three AI technologies in flash droughts detection. All these three metrics indices range between 0 and 1. POD and CSI show the ratio of detected flash droughts by the AI technologies to observed flash droughts, and the higher values, the better performances of AI technologies in flash droughts detection. FAR reflects the ratio of detected flash droughts that not occur in observations, with an optimal value of 0. These evaluation metrics can be expressed as follows:

$$POD = \frac{H}{H+M}, \quad (14)$$

$$FAR = \frac{F}{H+F}, \quad (15)$$

$$CSI = \frac{H}{H+F+M}, \quad (16)$$

where H (Hits) represents flash droughts both detected by the AI methods and observations; F (False alarms) represents the case when flash droughts captured by AI approaches but not recorded in observations. M (Misses)

represents flash droughts recorded in observations but not captured by AI approaches.





### 3.6 General framework

The general flowchart for evaluating the performances of AI technologies (i.e., MLR, LSTM, and RF model) in flash drought detection is presented in Fig. 2. We used a global reanalysis soil moisture dataset (i.e., ERA-Interim SM) to identify drought events and calculate their RI. Also, nine climate variables (i.e., P, PET, $T_{mean}$, $T_{min}$, $T_{max}$, RHU, PRS,

SSD, and WIN) collected from the in-situ observations were generated into spatially consistent climate element series by the IDW method. The process for flash droughts identification includes the following steps. Firstly, the original time series of these data were aggregated into weekly series, and the SM data were further transferred into the SM percentile based on the optimal selection of theoretical probability distribution function (PDF). Then, flash droughts were identified with a quantitative method by focusing on the intensification rate of soil moisture. The derived RI and

corresponding climate anomalies in the adjacent weeks of drought onset were serves as inputs to train MLR, LSTM, and RF models, respectively. Specifically, approximately 80% of drought events in each grid cell over China were applied to train the models, while the remaining drought events were used to test the performance of trained model. Finally, we evaluated the performances of the MLR, LSTM, and RF models by comparing the accuracies of RI simulation, the capabilities of flash droughts detection, and conducting a specific investigation on the typical drought

events.







**Figure 2:** The flow chart of evaluating the performances of AI models for flash droughts detection.

## 4 Results

### 4.1 Evaluation of the intensification rete of soil moisture

The capabilities of AI technologies in simulating the RI of soil moisture were assessed through intercomparison with the observed RI derived from ERA soil moisture. As shown in Fig. 3, higher RIs (up to 12.5th percentile per week for certain areas) were mostly concentrated in the southern part of China, e.g., the east of QTP, the east of SW, and the





middle and the south of MLYR regions. In contrast, lower RIs (less than 5.0th percentile per week for some regions) were mainly distributed in the southern XJ and western NW areas. Given the spatial heterogeneity of soil moisture,

Figs. 3b and c show the boxplots of RI in different sub-regions, as well as the changes of empirical cumulative distribution function (ECDF) of RI. It can be seen that the lowest RIs were mostly located in the XJ region, with the median value of 6.7th percentile per week. The highest RIs were distributed in the SW region with the median value of 12.5th percentile per week.

Based on the observed RI and simulations from three AI technologies (i.e., MLR, LSTM, and RF), Fig. 4 shows the

spatial distribution of CC and RMSE for the estimated RI against the observed RI in the testing phases during 1979-2016. For most parts of NE, SE and MLYR regions, there was generally a good agreement between the MLR simulated RI and observed RI with average CC values above 0.6, and average RMSE values below 5.0th percentile per week (Figs. 4a and b). The weaker correlations were mainly distributed in the southern part of XJ, as well as northern and western QTP. A similar spatial pattern was also found for LSTM simulated RI, but with overall boosted

consistency (Figs. 4c and d). Among the three AI models, the RF performed best, as shown in Figs. 5e and f, average CC values between RF simulated RI and the observed RI in most areas of China were more than 0.8, and the average RMSE are less than 4.0th percentile per week. Especially, the excellent estimations were found in the SE region with an average CC of 0.90 and average RMSE of 2.6th percentile per week, while the unsatisfying results were located in the XJ region with average CC of 0.75 and average RMSE of 3.3th percentile per week.

Figure 5 presents the ECDF of four evaluation coefficients (i.e., CC, RMSE, ME, and BIAS) of the estimated RI against the observed RI for all grids in China. It can be seen that the ECDF of CC and RMSE derived from the MLR and LSTM models were close to each other in different percentile intervals (Figs. 5a and b), as for ME and BIAS, the LSTM presented better estimations given the lower values of ME and BIAS (Figs. 5c and d). As for the RF model, the CC values were much higher than those of MLR and LSTM models, combined with lower values in terms of RMSE,

ME, and BIAS. Above analysis suggests that for RI estimation, the RF model was superior to MLR and LSTM model.



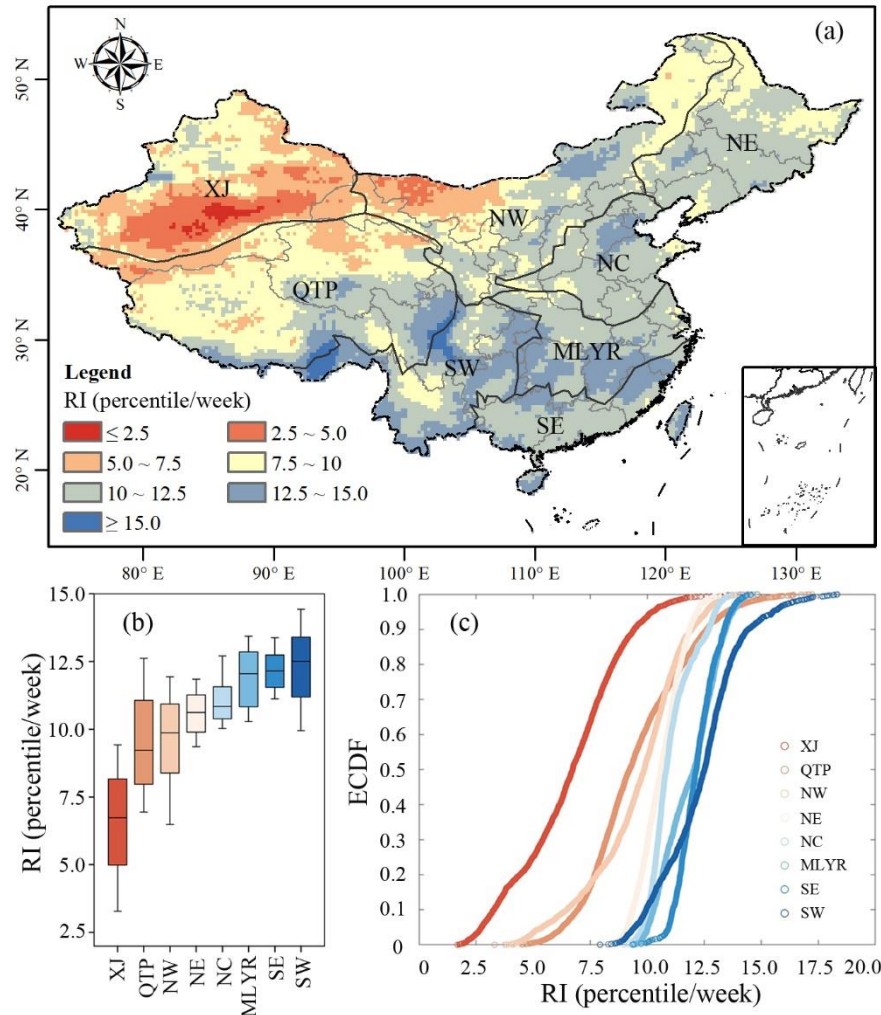

**Figure 3:** (a) Spatial distribution of the average rate of intensification (RI) during 1979-2016. (b) Boxplots of the average RI and (c) Empirical Cumulative probability distribution function (ECDF) of the average RI over different sub-regions in China. The sub-regions are Northeast China (NE), Northern China (NC), the middle and lower reaches of the Yangtze River regions (MLYR), Southeastern China (SE), Northwestern China (NW), Southwestern China (SW), Qinghai-Tibet Plateau (QTP), and Xinjiang (XJ).



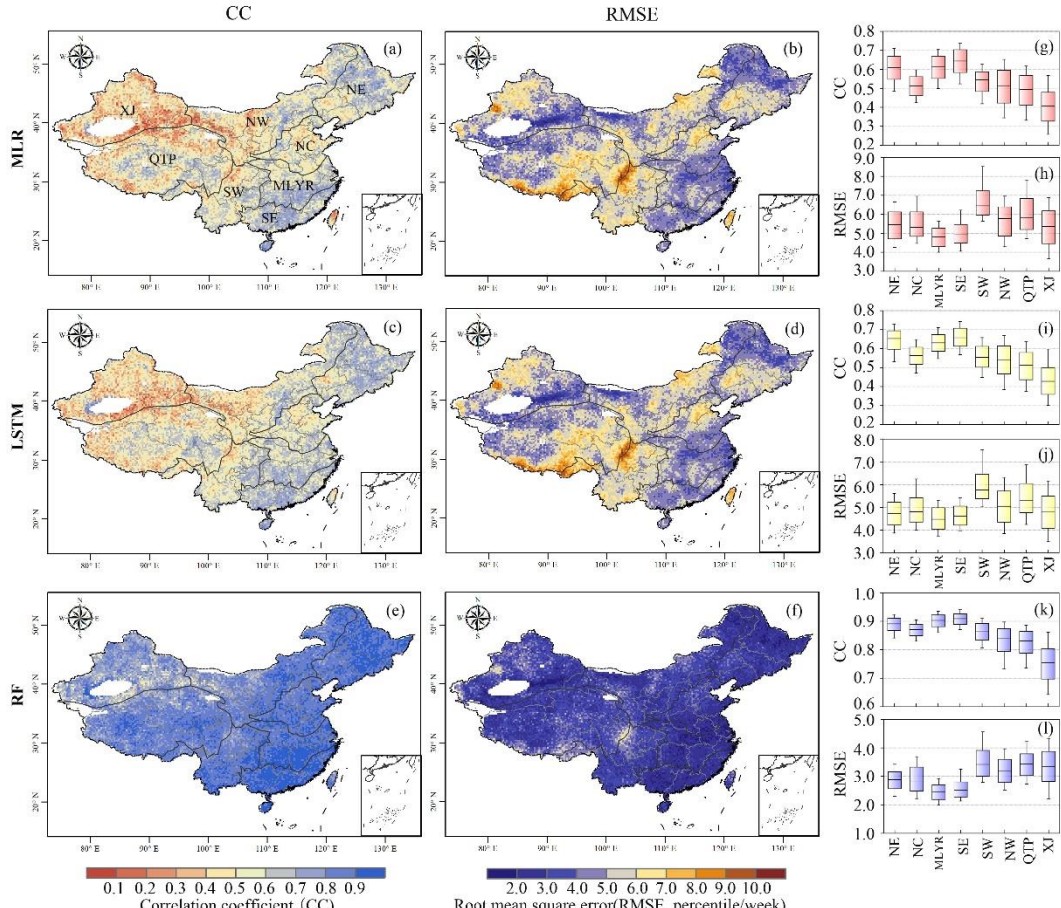

**Figure 4:** Spatial distribution of correlation coefficient (CC) and root mean square error (RMSE) of the estimated RI by (a-b) MLR, (c-d) LSTM, and (e-f) RF models against the observed RI. (g-l) Boxplots of the CC and RMSE for 320  eight sub-regions, respectively.

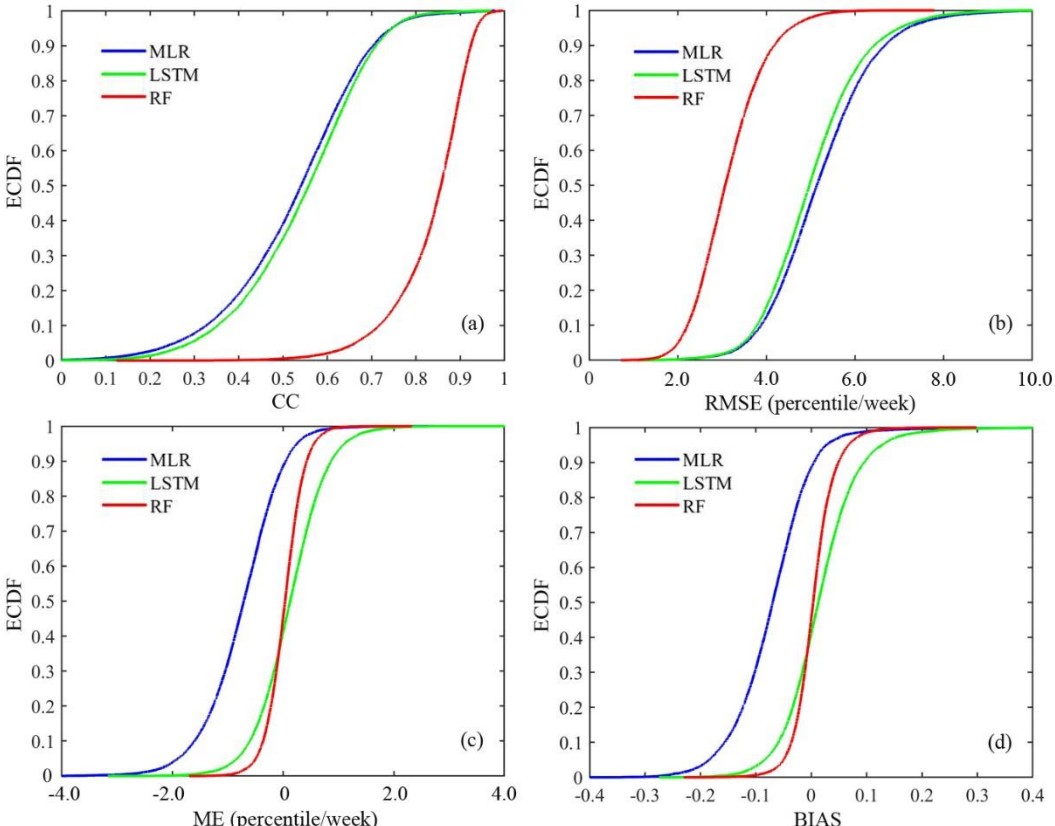

**Figure 5:** Empirical cumulative distribution function (ECDF) for (a) correlation coefficient (CC), (b) root mean squared error (RMSE), (c) mean error (ME), and (d) relative bias (BIAS) of the RI estimated by the MLR, LSTM, and RF models against observed RI.


## 4.2 Comparison of the RI of flash droughts and slowly-evolving droughts

RI is an important metric for distinguishing flash droughts from traditional slowly-evolving droughts. To evaluate the capabilities of three AI models in detecting drought events, we analyzed the correlation between model simulated RI and observed RI for flash droughts and slowly-evolving droughts, respectively (Fig. 6). For flash droughts, the MLR

and LSTM models displayed a similar spatial pattern, where higher CC values (up to 0.6 for some areas) were mainly located in the MLYR and SE regions, and correlations in the XJ and NW districts were generally weak (Figs. 6a and c). As for the RF model, except for some parts of the XJ region, it presented a rather high consistency with observed RI (CC values reach up to 0.9) in most areas of China. As for the case of traditional slowly-evolving droughts, the MLR and LSTM methods showed a weak correlation over whole China (Figs. 6b and d). With respect to the RF model, the





CC values overall increased in comparison with MLR and LSTM, with significant changes (the CC values increased by approximately 0.4) in SE and SW regions.

Figure 7 further exhibits the absolute errors and relative errors between the estimated RI and observed RI at four percentile intervals: 0th~5th, 5th~10th, 10th~15th, 15th~20th percentile per week. According to the flash drought identification method aforementioned, RI below the 5th percentile can be viewed as traditional droughts, while above
the 10th percentile were classified as flash droughts. For flash droughts, the good (absolute error below 1.0th percentile per week) performance of MLR was observed in NE, SE, SW, and MLYR regions, while the unsatisfying results were found in XJ and NW areas (Figs. 7j-l). As for the slowly-evolving droughts, the higher estimated accuracy (absolute error below 1.0th percentile per week) were mainly concentrated in XJ and NW regions, however, the unsatisfying results (absolute error over 10th percentile per week) were mostly located in the SW region (Figs. 7a-c). Besides, a
satisfactory estimation of RI with the value range 5th~10th and 10th~15th percentile per week were presented in most parts of China. Based on the above analysis, it can be concluded the MLR, LSTM, and RF algorithms can well simulate RI derived by flash drought in the NE, SE, SW, and MLYR regions, while these methods displayed a good estimation accuracy of RI indicated by traditional drought in XJ and NW regions.

Based on above analysis, we further evaluated the capabilities of the MLR, LSTM, and RF models for capturing flash
drought events and slowly-evolving drought events in eight different sub-regions by using three skill scores (i.e., POD, FAR, and CSI) (Fig. 8). For flash droughts, the average POD (FAR) of the MLR and LSTM models ranged from 0.58 to 0.88 (0.08 to 0.41) and 0.68 to 0.94 (0.10 to 0.44), respectively, which were much lower (higher) than those of the RF algorithm (Figs. 8a and b). Likewise, the CSI of the MLR and LSTM models were much lower than that of the RF methods. Figs. 8c and d present the cases of slowly-evolving droughts. It can be seen that the POD (FAR) of the MLR
(LSTM) model ranged from 0.41 to 0.70 (0.32 to 0.71), and 0.27 to 0.61 (0.30 to 0.66), respectively, while the values of RF approach varied from 0.34 to 0.72 (0.11 to 0.19). In terms of the CSI, the MLR and LSTM presented unsatisfying performances comparing to the RF model, with average values of 0.34, 0.30, and 0.51, respectively. Spatially, with the highest POD and CSI scores and the lowest FAR scores, the SE region exhibited the best detection results, and poor performances were in the XJ region. In general, all three AI models provided more reliable
information in detecting flash droughts than slowly-evolving droughts. Meanwhile, the RF is more recommended for use given its high skill scores and low false alarms in drought detection.





**Figure 6:** Spatial distribution of correlation coefficient (CC) of the rate of intensification (RI) estimated by (a-b) MLR, (c-d) LSTM, and (e-f) RF models against the observed RI under flash droughts and slowly-evolving droughts.





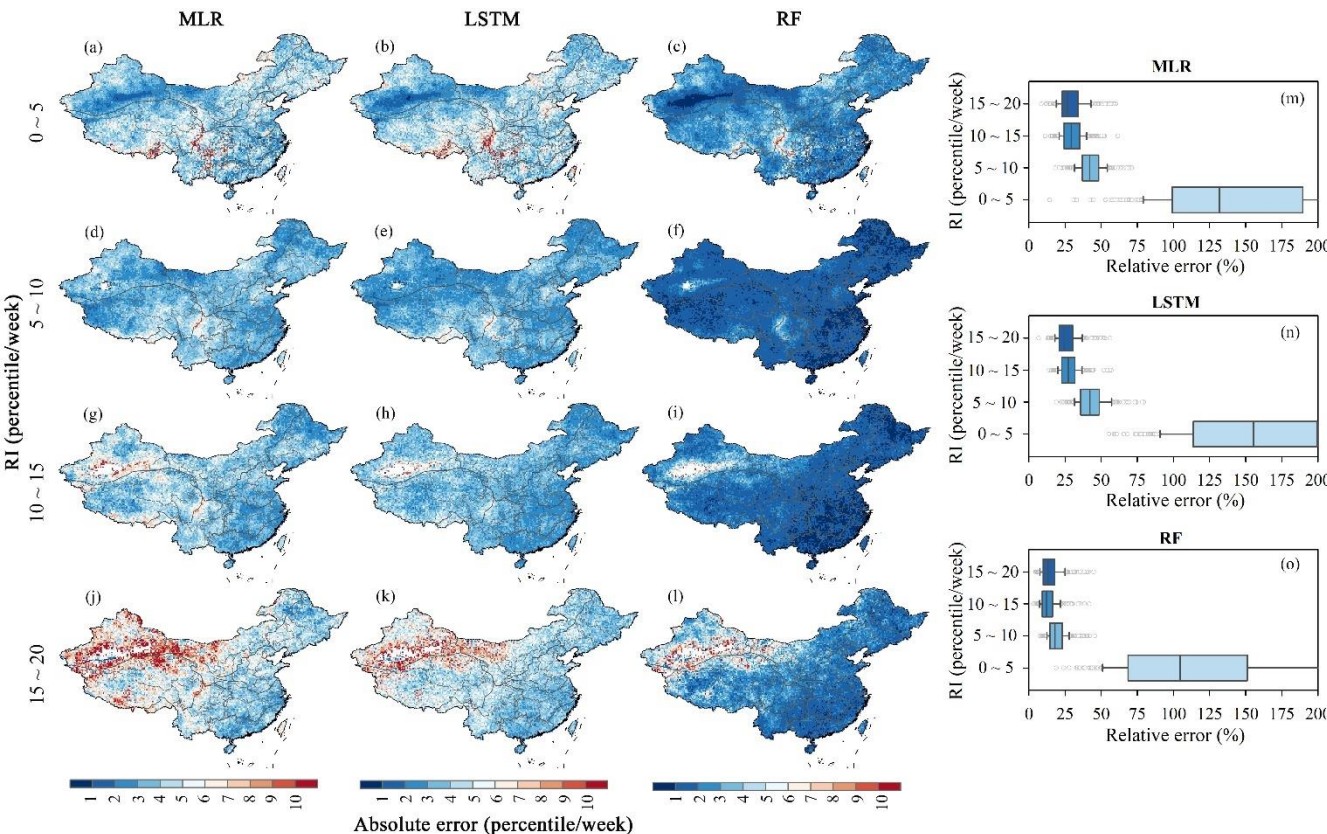


**Figure 7:** Spatial pattern of the absolute errors and boxplots of the relative errors of the RI estimated by the MLR, LSTM, and RF methods against observed RI at four percentile intervals.

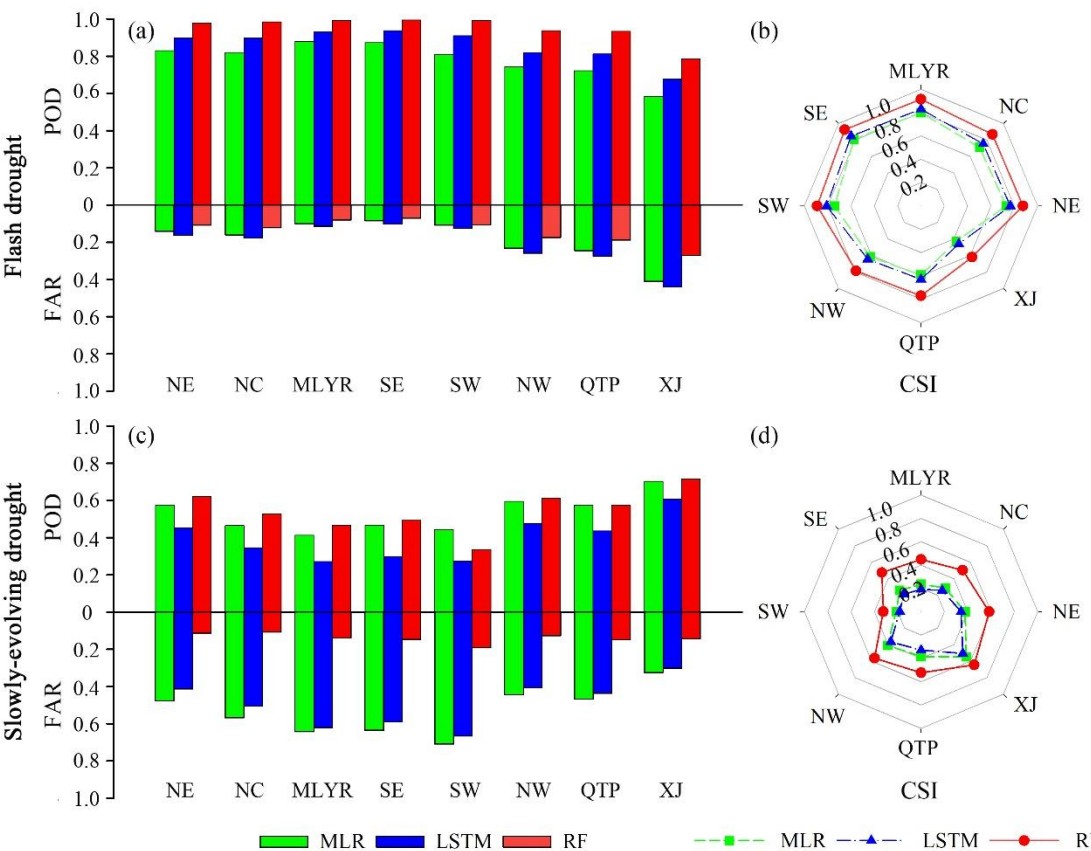

**Figure 8:** Skill scores of POD, FAR, and CSI for flash droughts and slowly-evolving droughts based on the MLR, LSTM, and RF models in eight sub-regions.

### 4.3 Spatiotemporal evolution of typical flash drought events

The ability of capturing the migration trajectories of droughts over time and space is also important for evaluating the capabilities of candidate AI models in drought detection. Fig. 9a displays the time series of flash drought area and slowly-developing drought area derived from ERA-Interim SM data during 1979~2016. As expected, the areas of slowly-evolving droughts overwhelmingly exceeded those of flash droughts, and the areal gaps were further enlarged after 2003. Figs. 9b and c exhibits the weekly variation of drought area in 2006 (the largest flash drought during the past 38 years with 11.73% of the area affected) and 2013. It can be seen that summer and autumn were two major seasons that the area of flash droughts and slowly-evolving drought developed towards different directions (increase or decrease). Given this, a specific investigation on the behaviors of MLR, LSTM, and RF in the summer and autumn of 2006 and 2013 was conducted to explore the capacities of the three AI models in monitoring the spatiotemporal migration trajectories of flash droughts. Figure 10 shows the spatial distribution of soil moisture percentile (first





column), observed RI (second column), and simulated RI by AI models (third to fifth column). From the perspective of the observed RI, the summer flash droughts mainly hit the NW, NC, SW and MLYR regions of China on 17 June, 2006 (Fig. 10b). Then the signal of flash droughts migrated towards the NE and SE regions on 23 September (Figs.

10g and l). Similarly, the 2013 summer flash droughts were mostly concentrated in MLYR areas with the average RI of 15.2th percentile per week (Fig. 10q). After 12 weeks, the flash droughts occurred on 17 October, and were mainly located in the SW area (Figs. 10v and aa). In terms of the accuracy of RI simulation, the MLR-estimated RI was generally higher than the observed RI in the SE and SW regions (Figs. 10h, m, w and ab). Comparing to the MLR algorithm, the simulated RI by the LSTM and RF approaches basically followed a nearly consistent pattern as the

observed RI, suggesting that they were superior to MLR in monitoring flash droughts.

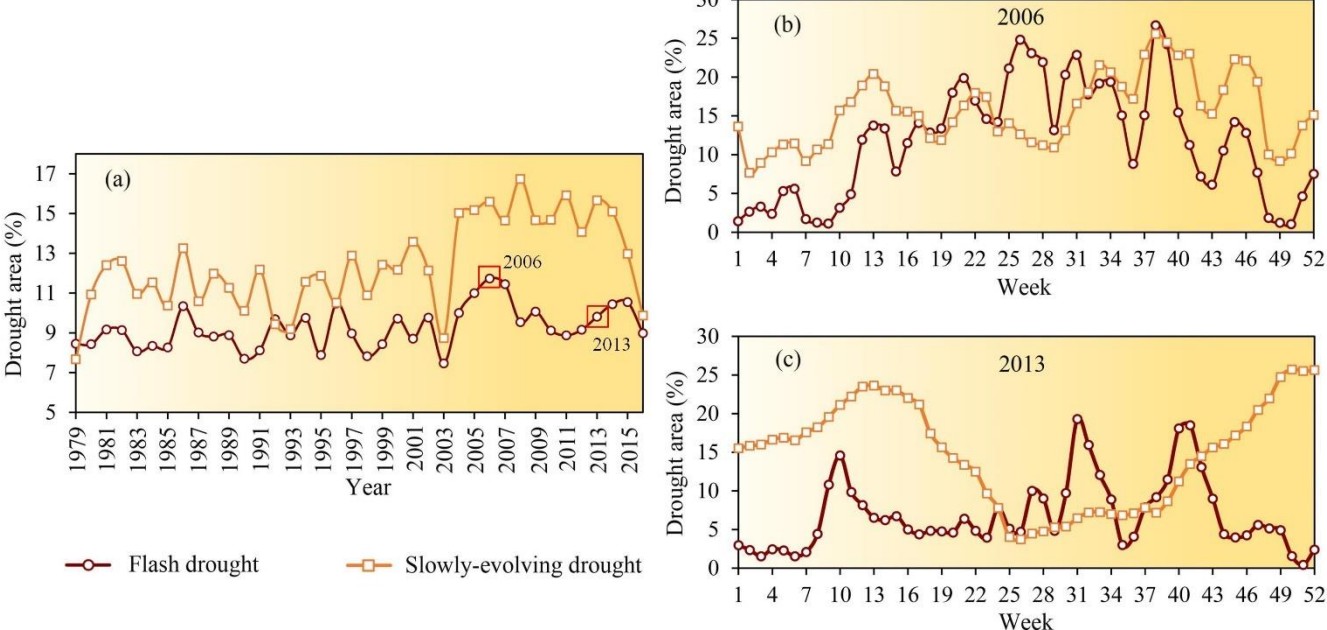

**Figure 9:** Time series of flash drought area and slowly-evolving drought area derived from ERA SM series during 1979~2016, as well as in the typical years of 2006 and 2013.

**Figure 10:** Spatial evolution of the weekly soil moisture percentile, the observed RI, and estimated RI from MLR, LSTM, and RF models over China in summer and autumn of 2006 and 2013.

## 5 Discussion

### 5.1 Performance of AI technologies for RI estimation

In this study, we evaluated three AI technologies, and found RF provided the best estimations of RI with higher CC and lower RMSE comparing to the observed RI (Figs. 4 and 5). It is not surprising that MLR did not perform well given its simple linear regression scheme which is insufficient to describe the complicated nonlinear relationships of variables. With complicated model structures, the LSTM performed slightly better than MLR, but its efficiency is not





optimistic either given the time-consuming calculations of the model. One possible reason lies in that the model requires the input and output data to have the same time step. In this study, the output of RI actually reflected the

average depletion rate of soil moisture during the onset-development stage, leading to the inconsistent temporal steps between output and input (i.e., meteorological variables), as mentioned in runoff prediction modeling (Xiang et al., 2020). Several previous studies also found the good behaviors of RF in constructing the nonlinear interactions between soil moisture and different land surface variables, and its strong capabilities for capturing the spatiotemporal variability of soil moisture (Zhao et al., 2017). For example, Fatholoumi et al. (2020) found due to the strong capability of

considering the complex linear and nonlinear relationships between soil moisture and land surface properties, RF outperform the MLR, Triangle regression, Inverse Distance Weighting, and Ordinary kriging techniques in estimating the variation of soil moisture in a semi-arid mountainous region. Rahmati et al. (2020) found the RF had excellent performances in mapping the agricultural drought hazards comparing to other machine learning technologies, including the classification and regression trees, boosted regression trees, multivariate adaptive regression splines,

flexible discriminant analysis, and support vector machines. The outstanding performance of RF could be attributed to the mathematical algorithm of the model which enables high classification accuracy, unbiased determination of generation error with the out-of-bag method, and high efficiency in extracting important information from complicated nonlinear interactions of variables in handling high-dimensional datasets (Naghibi et al., 2016; Rodriguez-Galiano et al., 2012; Wang et al., 2015).

Regarding the spatial heterogeneity of RI, we found the RF performed best in the southern China, while the estimation errors were high in the XJ region. This might be related to the local climate and soil conditions. Fig. 11 compares the variation of soil moisture, moisture-related (i.e., P and RHU) and energy-related (i.e., PET, $T_{mean}$, $T_{max}$, $T_{min}$, PRS, SSD, and WIN) meteorological factors in adjacent weeks (i.e., $T_{0-7}\sim T_{0+7}$) of the onset of drought events during 1979~2016 in XJ and SW regions of China. The XJ region is climatically drier with relatively thick soil layers and sparse vegetation,

and this climate and underlying surface conditions may be not beneficial to induce a rapid response of soil moisture to meteorological anomalies. From Figs. 11a, c, and e, we can see that for the XJ region, the variation of soil moisture was not consistent with the changes of meteorological anomalies for flash droughts. The sharp decline of soil moisture (with the value changing from 55.05th to 8.87th percentile within 2 weeks) in Fig. 11a is a typically rapid rate of intensification for flash droughts. However, the meteorological variables did not change synchronously, and even

presented lagging variations (e.g., P, PET, and $T_{mean}$) after the onset of flash drought. By contrast, the consistency between soil moisture and meteorological variables was considerably improved for slowly-evolving droughts (Figs. 11b, d, and f). As expected, the consistency degree was generally high in the SW region, with better behaviors for flash droughts. As shown in Fig. 11g, soil moisture decreased from 55.25th to 10.54th percentile within two weeks. Regarding meteorological variables, both P and RHU showed relatively stable negative anomalies (e.g., the value of P





anomaly and RHU anomaly at $T_0$ was -0.43, and -0.69, respectively), and energy-related variables (e.g., PET, T, WIN) presented continuously positive anomalies (e.g., the value of $T_{mean}$ anomaly and PET anomaly at $T_0$ was 0.28 and 0.59, respectively). All these contribute to the rapid decline of soil moisture. Different from the XJ region, the SW region belongs to humid climate zones with abundant soil moisture from the top to deep layers, accompanied with dense vegetation and well-developed root systems. In the joint effects of P deficit and high temperature or heat wave (Figs.

11g, i, and k), the capacity of evapotranspiration from vegetation could be enhanced in very short time period, leading to rapid response of soil moisture to the unusual climate conditions.



**Figure 11:** Time series of weekly soil moisture percentile, moisture-related (i.e., P and RHU), and energy-related (i.e., PET, $T_{mean}$, $T_{max}$, $T_{min}$, PRS, SSD, and WIN) climate factors in the adjacent week ($T_{0-7} \sim T_{0+7}$) of drought onset during 1979-2016 for flash droughts and slowly developing droughts in the XJ and SW regions. The blue shadows (Figs. 11a,





b, g, and h) denote the 25th~75th percentile range of soil moisture values. The dark yellow shadows in all 12 panels represent the onset-development phase of drought.

## 5.2 Comparison of AI technologies for flash droughts and slowly-evolving droughts

In this study, all three AI models produced better RI estimations of flash droughts than those of conventional droughts
(Figs. 6 and 8), suggesting that they are more competent to monitor the rapid onset of droughts. From the perspective of physical mechanisms, the formation of traditional slowly-evolving droughts commonly take a rather long time (e.g., several months or years) and they are driven by a variety of meteorological factors (Mishra and Singh, 2010). Precipitation deficits, enhanced evaporative demand, their joint or alternant effects are all possible to impose cumulative effects on soil moisture. Given the different climate and underlying conditions, the response time of the
hydrological system can be different, manifested as varied time scales of droughts (Zhu et al., 2021). Particularly, the driving forces of slowly-evolving droughts could be more diverse when considering the abnormal atmospheric circulation, which is the origin of meteorological droughts and is also responsible for soil moisture drought. For example, several previous studies suggest droughts essentially are resulted from the sea- and land-atmosphere interactions, and large-scale circulation factors such as the surface sea temperature, 500hPa geopotential height, and
850hPa vertical velocity all influence the development of drought (Xiao et al., 2016; Zeng et al., 2019). In other words, the complicated driving forces of slowly-evolving droughts at varying time scales make it difficult to simulate the variation of soil moisture from a climatic perspective.

In a different manner, flash drought particularly refers to the time period that rapid depletion of soil moisture occurs (Otkin et al., 2018), which usually requires the simultaneous anomalies in precipitation, relative humidity, potential
evapotranspiration, temperature, sunshine duration, wind speed, and other meteorological variables to integrate into strong climatic forces (Liu et al., 2020a; Hobbins et al., 2016; Hunt et al., 2014). This rigorous atmospheric driving condition theoretically would not sustain for a long time, and a pentad or weekly time scale is recommended for monitoring flash droughts. Comparison on the individual roles of precipitation (representing the water supply condition) and PET (representing the limits of evaporative demand) in formulating flash droughts and slowly-evolving
droughts also showed this difference. Taking the case of the MLR method as an example, Fig. 12 exhibits the weights of P and PET anomalies in the adjacent weeks (as $T_{0-7}\sim T_{0+7}$ in Fig. 11) of drought onset for the XJ and SW regions. It can be seen that the weights of P and PET anomalies for flash droughts were generally higher than those of traditional droughts, suggesting a closer relationship between meteorological variables (i.e., P and PET) and flash droughts. Meanwhile, regional differences associated with the individual roles of P and PET were also observed. For the XJ
region, the weights of negative P anomaly were generally high at the beginning of two types drought, while the maximum weight of the positive PET anomaly occurred almost after drought onset. As for the SW region, both the



negative P anomalies and the positive PET anomalies presented high weights during the onset time of droughts. The results suggested that P deficit played an important role during drought onset in the XJ region, and for the SW region, the lack of precipitation and elevated evaporative demand both played important roles for the occurrence of droughts,

and this synchronously combined effect on the depletion of soil moisture is particularly significant for flash droughts. In general, the AI models are more competent to capture the variation of RI for flash droughts than the slowly-evolving drought due to the close causative relationship between meteorological forces and the former drought type.

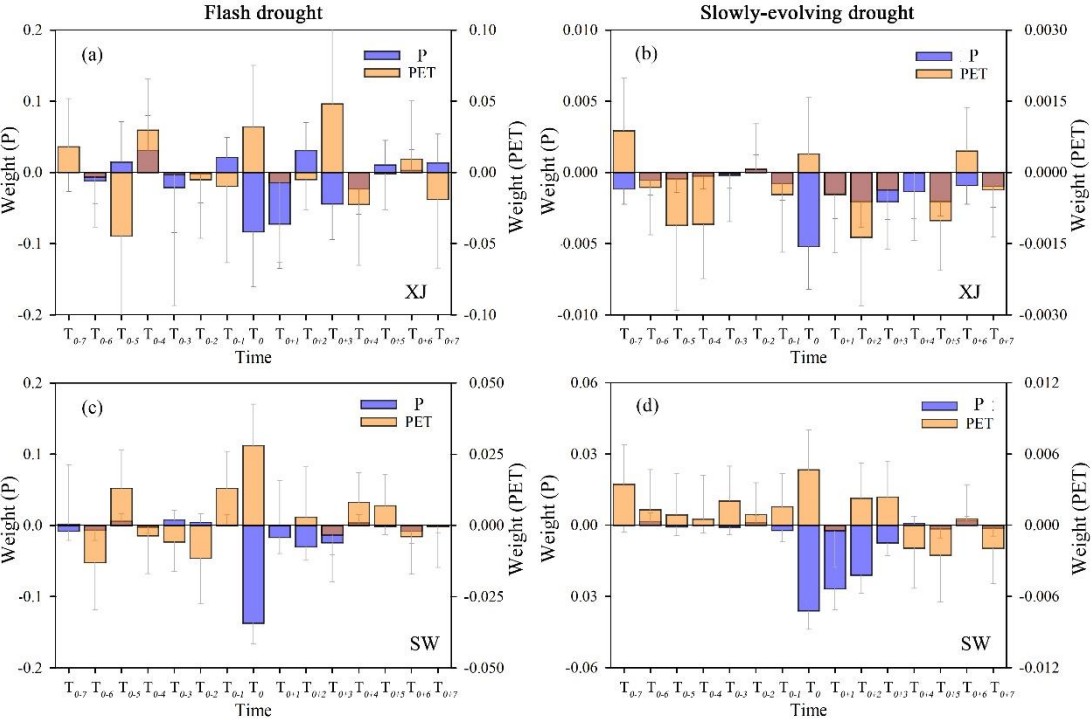

**Figure 12:** The weights of P (blue bar) and PET (yellow bar) for flash droughts and slowly-evolving droughts based
on MLR method in adjacent weeks of drought onset in the XJ and SW regions. $T_{0-1}$ denotes 1-week prior to the onset time, while $T_{0+1}$ represent 1-week after the onset time.

## 6 Conclusions

Based on the depletion rate of soil moisture derived from the ERA-Interim dataset, we identified flash droughts across China during 1979~2016. Furthermore, the linear and nonlinear relationships between ERA-Interim soil moisture and
multiple climate variables were constructed using the MLR, LSTM, and RF technologies. On this basis, we evaluated the performance of these models in estimating the rate of intensification (RI) of soil moisture and analyze their capabilities on flash drought detection. Overall, the RF model displayed the best performance for the whole of China,





which was much better than that of MLR and LSTM models. The highest results estimated by RF were in the NE region, with an average CC of 0.90 and average RMSE of 2.6th percentile per week, while the lowest estimations were found in the XJ area, with average CC of 0.75 and average RMSE of 3.3th percentile per week. A specific investigation on the summer and autumn droughts in 2006 and 2013 indicated that RF and LSTM can well reveal the spatial patterns of RI. They were able to provide a better simulation of flash drought relative to MLR with the lowest estimations. Furthermore, these AI methods displayed a relatively higher detection capacity of flash droughts than that of traditional slowly evolving droughts. RF model was recommended to simulate flash drought by considering the multiple meteorological variable anomalies in the adjacent time period of drought onset. The POD, FAR, and CSI of flash drought captured by the RF were 0.93, 0.15, and 0.80, respectively. In terms of the meteorological driving mechanism of flash droughts, the negative precipitation (P) anomalies and positive potential evapotranspiration (PET) anomalies exhibited a stronger synergistic effect on flash droughts comparing to slowly-developing droughts. Such compound effects on flash drought also presented asymmetrical characteristics over two regions in China. For the XJ region, P deficit played a dominant role on driving the onset of droughts, while for the SW region, the lack of precipitation and elevated evaporative demand contributed almost equally for the occurrence of droughts. This work would help enhance the understanding of flash droughts and provide a reference for the application of AI models on simulating flash droughts.

### Declaration of Competing Interest

The authors declare that they have no known competing financial interests or personal relationships that could have appeared to influence the work reported in this paper.

### Acknowledgments

This study was supported by the National Key Research and Development Program approved by the Ministry of Science and Technology, the People's Republic of China, under Grant No. 2019YFC1510600; the Fundamental Research Funds for the Central Universities under Grant No. B200204029, No. B200203054; the National Science Foundation of China under Grant No. 42171021, No. 41901037, and No. 42071040; the Fundamental Research Funds for the Central Universities under Grant No. 2019B05214; the Postgraduate Research & Practice Innovation Program of Jiangsu Province, under Grant No. KYCX20_0468; the Central guidance for local science and technology development funds projects under Grant No. 2021ZY0027.



## Data availability statement

ERA-Interim SM data are available through https://apps.ecmwf.int/datasets/data/interim-full-daily/levtype = sfc/.
Meteorological observation records are available from the China Meteorological Administration website (CMA,
http://data.cma.cn/).

**Author Contribution**

L. Zhang carried out the analyses, wrote the manuscript, and prepared the figures. Y. Liu and L. Ren designed the
paper and supervised the formulation of this manuscript. A.J. Teuling and Y. Zhu provided critical feedback and edits.
L. Wei and L. Zhang prepared the data. S. Jiang, X. Yang, X. Fang, and H. Yin provided important suggestions. All
authors discussed the results and contributed to the final manuscript.

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
