# Peer review of "Analysis of Flash Droughts in China using Machine Learning"

_Hydrology and Earth System Sciences, 2021_

## Author Comment (AC1)

Response to reviewer #1

1. This paper studies the predictability of flash drought over China using machine learning methods. The starting point is ERA5 soil moisture over China for the period 1979-2021. They use a definition of flash drought based on changes in soil moisture percentiles (SMP) which they term the rate of intensification (RI) during periods when SMP is decreasing. They define flash droughts as occurring when SMP crosses the 40th percentile and is decreasing at a rate of at least 6.5 percent per week (time step is weekly). There is some confusion in Figure 1 and text surrounding it as to whether crossing of the 20th percentile of SMP is also required (the figure implies this, but text does not). There also is a criterion for a termination time Tn "when the rapid decline of soil moisture ceases", but this is not shown in Figure 1 nor are specifics in the text.

Response: We thank the reviewer's comment. In the revised manuscript, we added some descriptions in Page 5 Lines 159-160 to clarify two detailed requirements for extracting drought events: "*Specially, the drought events are extracted by following two requirements below: (1) soil moisture falls below the 40th percentile, and (2) there exists the time period when soil moisture is less than the 20th percentile.*". The original symbol $T_n$ which represents the termination time of the onset-development phase was replaced by $T_{0+d}$ in order to keep its consistency between Fig. 1(a) and Fig. 1(b). We adjusted the formula of the intensification rate of drought events and revised some related descriptions as below:

In Page 6 Lines 174-178:

$$RI = \frac{1}{d+1}\sum_{i=0}^{d}\left[\frac{SM(T_{i+1})-SM(T_i)}{T_{i+1}-T_i}\right], \quad T_0 \leq T_i \leq T_{0+d}, \tag{1}$$

$$s.t = \{\min[SM(T_i)] \leq 20th\}, \tag{2}$$

"*Where $T_0$ is the onset time, $T_{0+d}$ denotes the termination time for the onset-development phase, d is the duration of onset-development phase, $SM(T_i)$ is the soil moisture percentile at time $T_i$ in the rapid intensification process of drought.*"

In Page 6 Lines 168-171:

"*$T_{0+d}$ denotes the termination time for the onset-development stage when the rapid decline of soil moisture ceases…*"

"*$T_{0+d}$ can be determined through a polynomial function and located when the first derivative of the constructed polynomial equals zero in calculus.*"

Besides, we revised Figure 1 in the original manuscript by showing the termination time $T_{0+d}$ clearly. For the specific identification method of the termination time, we

added a sentence in Page 6 Lines 171: *"The detailed determination process of $T_{0+d}$ is presented in our previous study (Liu et al. 2020a)."*

[Figure]

Fig.1 A concept map for identifying flash droughts.

2. My main problem with this paper is philosophical. Why are you using machine learning at all? It reflects no physical process understanding water – you just throw a bunch of variables that you think could possibly have something to do with RI and turn the crank. Rather obviously, flash droughts are going to occur during dry periods (during precipitating periods, presumably soil moisture increases rather than decreases). So given that it's dry, it must have to do with evaporative demand, and the soil moisture you start with. We do understand those processes (albeit imperfectly), so surely you could use a physically based model to predict the RI. Now, if you did that first, and then applied ML and could somehow (not clear at all to me how) use the ML predictions to diagnose the physically based ones so as to improve them, I would be interested. But I don't really see where the hydrologic content is in this paper.

**Response:**

Thanks for your comments. We agree that a physically-based model is helpful to understand the physical process of flash drought. Flash drought is an emerging and ongoing topic in the drought community during the past ten years. It has a rather

complicated evolving process and is influenced by a variety of factors in its different development stages. A variety of studies have analyzed the characteristics of flash drought at global and regional scales, though there is no consistent notion on how we define flash droughts in the community (e.g., Ford et al. 2015; Otkin et al. 2018; Liu et al., 2020). Especially, the physically driven mechanism of flash drought is still uncertain. Given the current progresses of the flash drought field, it is difficult to predict RI using physical-based models. The three machine learning (ML) technologies (i.e., MLR, RF, and LSTM) were used to establish the relationship between RI and predictors, which can be served as references for recognizing and understanding the physical formation and development features of flash droughts.

We agree that it is difficult to use ML technologies to reflect the physical process of flash drought. However, these ML technologies have advantages in providing a fast and direct mapping pathway between the independent and dependent variables based on a combination of abundant data and advanced model architectures (Feng et al., 2021; Yang et al., 2020). Also, they are able to provide an accurate estimation of soil moisture, though the input samples are limited (Long et al., 2019). Given this, we considered using the ML models to evaluate the feasibility of flash droughts simulation over China. Our current study is to figure out the statistical interaction between RI and the anomalies of meteorological factors, which is beneficial for understanding the physical mechanism of flash droughts.

Besides, we appreciate the reviewer for providing a good idea that we can diagnose the physical-based models using machine learning technologies so as to improve the performance of the former models. This might be a study direction in the future flash drought field. However, considering the reason for the unclear physical mechanism of flash droughts, we applied the ML models to analyze the potential relationship between the RI and anomalies of climate factors in the current work. This is the first step before we effectively construct physical-based models to simulate and predict the RI in the future. The reason for using machine learning algorithms has been supplemented in the introduction section in Page 4 Lines 107-113:

"*These ML technologies have some advantages in providing a fast and direct mapping pathway between the independent and dependent variables based on a combination of abundant data and advanced model architectures (Feng et al., 2021; Yang et al.,*

*2020). Also, they are able to provide an accurate estimation of soil moisture, though the input samples are limited (Long et al., 2019). Given the inconsistent definition of flash drought and its uncertain driving mechanism, we applied the ML models to simulate flash droughts, which is beneficial for understanding their physical mechanism in the future. Meanwhile, limited studies focused on flash droughts based on ML technologies.*"

References:

Feng, Z., Niu, W., Tang, Z., Xu, Y., Zhang, H.: Evolutionary artificial intelligence model via cooperation search algorithm and extreme learning machine for multiple scales nonstationary hydrological time series prediction, J. Hydrol., 595, 126062, https://doi.org/10.1016/j.jhydrol.2021.126062, 2021.

Ford, T. W., McRoberts, D. B., Quiring, S. M., Hall, R. E.: On the utility of in situ soil moisture observations for flash drought early warning in Oklahoma, USA, Geophys. Res. Lett., 42:9790–9798, https://doi.org/10.1002/2015GL066600, 2015.

Liu, Y., Zhu, Y., Ren, L., Otkin, J., Jiang S.: Two Different Methods for Flash Drought Identification: Comparison of Their Strengths and Limitations, J. Hydrometeorol., 21: 691-704, https://doi.org/10.1175/JHM-D-19-0088.1, 2020.

Long, D., Bai, L., Yan, L., Zhang, C., Shi, C., Yang, W., Lei, H., Quan, J., Meng, X., Shi, C.: Generation of spatially complete and daily continuous surface soil moisture of high spatial resolution, Remote Sens. Environ., 233, 10.1016/j.rse.2019.111364, 2019.

Otkin, J. A., Svoboda, M., Hunt, E. D., Ford, T. W., Anderson, M. C., Hain, C., Basara, J. B.: Flash Droughts: A review and assessment of the challenges imposed by rapid onset droughts in the United States, Bull. Am. Meteorol. Soc., 99(5), 911-919, https://doi.org/10.1175/BAMS-D-17-0149.1, 2018.

Yang, S., Yang, D., Chen, J., Santisirisomboon, J., Zhao, B.: A physical process and machine learning combined hydrological model for daily streamflow simulations of large watersheds with limited observation data, J. Hydrol., 590(1): 125206, http://doi.org/10.1016/j.jhydrol.2020.125206, 2020.

3. My other complaint is that key information needed to understand the results is either buried in text or missing altogether. For instance, were flash drought periods extracted from the entire period of record, without regard for season? Ordinarily, one would expect such events to occur primarily in summer, when evaporative demand is the highest. But RI is determined in terms of soil moisture percentage changes, which complicates the picture considerably. In winter, for instance, evaporative demand will be reduced, but the range of soil moisture percentages likely is also reduced, so it could be that the statistics of RIs are being dominated by events that in a practical

sense aren't really droughts at all. I don't know if this is true but constraining the analysis to a window in the summer (if this hasn't already been done – I searched the document and didn't find any indication that it was) would make the most sense.

**Response**:

Thanks for pointing out that. Yes, we extracted flash drought from the entire period of record. When we designed the manuscript, we also first focused on flash droughts in the summertime as the reviewer suggested. We analyzed the identification results carefully and found flash droughts are prone to occur during the cross seasons (e.g., spring-summer or the summer-autumn). Constraining the analysis to a window in the summer may miss the continuous development process of flash drought. Given the above considerations, we preferred to analyze flash droughts by using the entire period. The main reasons are listed below: Firstly, our method relies on continuous time series of soil moisture percentile. The intermittent data makes it hard to capture the onset, or termination of drought events accurately, and the continuity and integrity of the datasets are important for identifying the development process of drought. Secondly, some important information related to flash droughts might be ignored if we merely focus on them in the summer. Previous studies showed that flash droughts may coexist with the seasonal drought and cross-seasonal drought due to the diverse climatic conditions and underlying surface (i.e., the soil texture and vegetation cover) of China (Liu et al., 2020). Meanwhile, according to their study, cross-season drought events easily started from spring (April and May) and summer (June and July). We also analyze the frequency of flash drought occurrence (FOC, Mo et al., 2016) in different seasons, as shown in Fig. 2. According to our identification results, the frequency of flash drought occurrence in winter is the lowest, and for 84% of the study area, the FOC is no more than 5% (Fig. 2a). This low value may have tiny influences on the simulation results. Based on the above analysis, we were more inclined to use the entire period for RI simulation and prediction of flash droughts. We considered your suggestion carefully and supplemented some expressions in Page 6 Lines 179-186 in the revised manuscript to explain the reason why we focused on the entire period.

*"We extracted flash drought from the entire period of record, the main reasons are listed: Firstly, our method relies on continuous time series of soil moisture percentile. The intermittent data makes it hard to capture the onset, or termination of drought*

*events accurately, and the continuity and integrity of the datasets are important for identifying the development process of drought. Secondly, enough important information related to flash droughts need be included in the ML models because flash droughts may coexist with the seasonal drought and cross-seasonal drought due to the diverse climatic conditions and underlying surface (i.e., the soil texture and vegetation cover) of China (Liu et al., 2020). Thirdly, the lower occurrence of flash drought in winter may have tiny influences on the simulation results."*

[Figure]

Fig.2 Spatial distribution of frequency of occurrence of flash droughts in different seasons over China

References:

Mo, K. C., Lettenmaier, D, P.: Precipitation deficit flash droughts over the United States, J. Hydrometeorol., 17(4): 1169-1184, https://doi.org/10.1175/JHM-D-15-0158.1, 2016.

Liu, Y., Zhu, Y., Zhang, L., Ren, L., Yuan, F., Yang, X., Jiang, S.: Flash droughts characterization over China: From a perspective of the rapid intensification rate, Sci. Total Environ., 704,135373, https://doi.org/10.1016/j.scitotenv.2019.135373, 2020b.

---

## Author Comment (AC2)

Response to reviewer #2

Overall, I consider this to be a worthwhile contribution to the rapidly expanding flash drought literature. The authors provide a new definition that can be compared to other proposed definitions and they examine association with a range of potential drought predictors. My two major comments are on the framing and the comparison between flash droughts and "slow droughts."

Response: We thank the reviewer for the positive comments to our study and please see our responses in detail below.

Major comments:

1. The methods applied in the study are, formally, supervised statistical learning algorithms. While one can debate what "AI" means, I think it's fair to assume that very few people think of linear regression, or even nonparametric statistical approaches like Random Forest, as AI. LTSM does sometimes get put in the AI basket, but it's no longer really a leading edge, advanced AI application. All that to say, I was surprised by the content of the manuscript after reading the title, and I suspect others may be as well. The paper simply does not provide an AI-oriented methodological advance, nor does it present results that are interesting because of novel application of relatively new methods. For this reason I recommend retitling and reframing the paper to focus on the flash drought findings, and removing the prominent use of the term AI in title, abstract, and throughout the paper. There are many published studies in many fields that compare performance of parametric and nonparametric methods for various applications, sometimes including NN as well, and at this point I really think that the difference in performance between those methods is best presented as a comparison of statistical methods that is useful but not particularly innovative. Instead, I recommend that the authors focus on their actual flash drought results in the framing of the paper, as those results are quite interesting for the flash drought community.

**Response:**

Thank you for pointing out that. We agree with the reviewer's comment that these three methods (i.e., MLR, RF, and LSTM) are inappropriate to consider as artificial intelligence (AI) technologies. As you mentioned, these parametric and nonparametric methods, and they were named as machine learning (ML) technologies in previous studies (Bouras et al., 2021; Liakos et al., 2018; Schwalbert et al., 2020). Following

your suggestions and previous studies, we classified these methods i.e., MLR, RF, and LSTM into machine learning technologies and modified the original title to "*Flash drought simulation based on machine learning technologies with time-adjacent meteorological conditions*". The new title would be better to reflect the key point of flash drought in this work. We corrected sentences containing AI terms and replaced them with descriptions of machine learning technologies. The detailed revisions are shown as below:

In Page 1 Lines 17-18:

[revised manuscript text omitted]

Reference:

Bouras, E. h., Jarlan, L., Er-Raki, S., Balaghi, R., Amazirh, A., Richard, B., Khabba, S.: Cereal
        Yield Forecasting with Satellite Drought-Based Indices, Weather Data and Regional Climate

Indices Using Machine Learning in Morocco, Remote Sens., 13, 3101, https://doi.org/10.3390/rs13163101, 2021.

Liakos, K. G., Busato, P., Moshou1, D., Pearson, S., Bochtis, D.: Machine Learning in Agriculture: A Review, Sensors, 18, 2674, http://doi.org/10.3390/s18082674, 2018.

Schwalbert, R. A., Amado, T., Corassa, G., Pott, L. P., Prasad, P. V. V., Ciampitti, I. A.: Satellite-based soybean yield forecast: Integrating machine learning and weather data for improving crop yield prediction in southern Brazil, Agric. For. Meteorol., 284, 107886, http://doi.org/10.1016/j.agrformet.2019.107886, 2020.

2. I appreciate the section of the manuscript that compares the predictability of flash drought to conventional drought. But in making this distinction the authors implicitly assume that flash and slow droughts, as distinguished using the RI threshold employed in this paper, are meaningful and relatively homogeneous types of drought with respect to the predictor variables. Are the flash droughts and slow droughts in the inventory relatively homogeneous and separable with respect to these predictors, when evaluated using standard clustering or homogeneity tests? And is there evidence of the greater spread in meteorological predictors for slow drought relative to flash drought, as the authors suggest when explaining poorer performance in predicting slow droughts as a function of meteorology?

**Response:**

We thank the reviewer for their work and the positive comments. Yes. Flash droughts and slowly-evolving droughts are relatively homogenous and separable with respect to these predictors. In this study, we first identified drought events and calculated the decline rate of soil moisture. Our identification method followed the suggestion of Otkin et al. (2018) and was similar to the previous literature (Yuan et al., 2017; Ford et al., 2015) which focused on two key characteristics of flash drought, namely the intensification rate to reflect how fast the drying status proceeds, and the upper (40th) and lower (20th) limits of soil percentile to guarantee the event really falls into drought. Then, the RI of different drought events (including both flash droughts and traditional slowly-evolving droughts, and flash droughts can be distinguished from conventional droughts based on the RI threshold of "-6.5th percentile/week"), as well as relevant predictors, were employed as inputs to the ML models. Finally, the feasibilities of flash drought and slowly-evolving drought simulation were evaluated.

Traditional drought is influenced by a variety of predictors actively involved in the

physical processes of the atmosphere, ocean, and land (Hao et al., 2018), which bring great challenges for the prediction of drought. These predictors can be divided into three types: (1) The first type of predictors is the large-scale climate indices, for instance, Surface Sea Temperature (SST), Southern Oscillation Index (SOI), Pacific Decadal Oscillation (PDO), and North Atlantic Oscillation (NAO). The large-scale teleconnection factors have been shown to be an important driving force for the occurrence and development of drought in different areas of the world (Hoerling et al., 2003; Nicolai-Shaw et al., 2016; Trambauer et al., 2013). (2) The second type of predictors refer to the local climate variables (e.g., precipitation, temperature). For example, under the joint effects of precipitation deficit and high temperature, soil moisture may be declined and persistent moisture deficits may lead to agricultural drought (Otkin et al., 2018; Yuan et al., 2019). (3) The land initial conditions (e.g., the persistence of soil moisture) can also be used as predictors for the prediction of drought (Wu et al., 2021). Especially for flash droughts, relevant studies showed that they have a stronger meteorological driving demand than conventional droughts (Ford and Labosier, 2017; Liu et al., 2021). This suggests a close interaction between RI and these local meteorological conditions, and this may be one reason for the relatively high efficiencies of these meteorological variables for RI prediction. By contrast, the formation of traditional drought involves complicated atmosphere-land surface feedbacks at multiple scales, and it is difficult to efficiently capture the variation of RI for slowly-evolving drought from a meteorological perspective.

---

## Author Response (AR1)

**Response to Editor and Reviewers**

**Manuscript ID HESS-2021-541**

**Journal:** *Hydrology and Earth System Sciences*

**Article title:** Analysis of flash droughts in China using machine learning technologies

**Author list:** *Linqi Zhang, Yi Liu, Liliang Ren, Adriaan J. Teuling, Ye Zhu, Linyong Wei, Linyan Zhang, Shanhu Jiang, Xiaoli Yang, Xiuqin Fang, Hang Yin*

Dear reviewers and editor:

Thank you so much for valuable comments and kind suggestions on our paper. Your illuminating comments and suggestions give us the possibility to properly fix several questionable issues, and to improve the overall quality of the paper. We highly appreciate your time and effort. Please find our point-to-point responses to your comments below.

**Response to reviewer #1**

1. This paper studies the predictability of flash drought over China using machine learning methods. The starting point is ERA5 soil moisture over China for the period 1979-2021. They use a definition of flash drought based on changes in soil moisture percentiles (SMP) which they term the rate of intensification (RI) during periods when SMP is decreasing. They define flash droughts as occurring when SMP crosses the 40th percentile and is decreasing at a rate of at least 6.5 percent per week (time step is weekly). There is some confusion in Figure 1 and text surrounding it as to whether crossing of the 20th percentile of SMP is also required (the figure implies this, but text does not). There also is a criterion for a termination time Tn "when the rapid decline of soil moisture ceases", but this is not shown in Figure 1 nor are specifics in the text.

Response: We thank the reviewer's comment. In the revised manuscript (no marks), we added some descriptions in Page 5 Lines 159-160 to clarify two detailed requirements for extracting drought events: "*Specifically, the drought events are extracted from the entire period by following two requirements below: (1) soil moisture falls below the 40th percentile, and (2) soil moisture should decay to below the 20th percentile.*". The original symbol $T_n$ which represents the termination time of

the onset-development phase was replaced by $T_{0+d}$ in order to keep its consistency between Fig. 1(a) and Fig. 1(b). We adjusted the formula of the intensification rate of drought events and revised some related descriptions as below:

In Page 6 Lines 173-177:

$$RI = \frac{1}{d+1}\sum_{i=0}^{d}\left[\frac{SM(T_{i+1})-SM(T_i)}{T_{i+1}-T_i}\right], \quad T_0 \leq T_i \leq T_{0+d}, \tag{1}$$

$$s.t = \{\min[SM(T_i)] \leq 20th\}, \tag{2}$$

*"Where $T_0$ is the onset time, $T_{0+d}$ denotes the termination time for the onset-development phase, d is the duration of onset-development phase, $SM(T_i)$ is the soil moisture percentile at time $T_i$ in the rapid intensification process of drought."*

In Page 6 Lines 168-170:

*"$T_{0+d}$ denotes the termination time for the onset-development stage when the rapid decline of soil moisture ceases…"*

*"$T_{0+d}$ can be determined through a polynomial function and located when the first derivative of the constructed polynomial equals zero in calculus."*

Besides, we revised Figure 1 in the original manuscript to show the termination time $T_{0+d}$ clearly. For the specific identification method of the termination time, we added one sentence in Page 6 Lines 170-171: *"The detailed determination process of $T_{0+d}$ is presented in our previous study (Liu et al. 2020a)."*

[Figure]

Fig.1 A concept map for identifying flash droughts.

2. My main problem with this paper is philosophical. Why are you using machine learning at all? It reflects no physical process understanding water – you just throw a bunch of variables that you think could possibly have something to do with RI and turn the crank. Rather obviously, flash droughts are going to occur during dry periods (during precipitating periods, presumably soil moisture increases rather than decreases). So given that it's dry, it must have to do with evaporative demand, and the soil moisture you start with. We do understand those processes (albeit imperfectly), so surely you could use a physically based model to predict the RI. Now, if you did that first, and then applied ML and could somehow (not clear at all to me how) use the ML predictions to diagnose the physically based ones so as to improve them, I would be interested. But I don't really see where the hydrologic content is in this paper.

**Response:**

Thanks for your comments. We agree that a physically-based model is helpful to understand the physical process of flash drought. Flash drought is an emerging and ongoing topic in the drought community during the past ten years. It has a rather complicated evolving process and is influenced by a variety of factors in its different development stages. A variety of studies have analyzed the characteristics of flash

drought at global and regional scales, though there is no consistent notion on how we define flash droughts in the community (e.g., Ford et al. 2015; Otkin et al. 2018; Liu et al., 2020a). Especially, the physically driven mechanism of flash drought is still uncertain. Given the current progresses of the flash drought field, it is difficult to predict RI using physical-based models. The three machine learning (ML) technologies (i.e., MLR, RF, and LSTM) were used to establish the relationship between RI and predictors, which can be served as references for recognizing and understanding the physical formation and development features of flash droughts.

We agree that it is difficult to use ML technologies to reflect the physical process of flash drought. However, these ML technologies have advantages in providing a fast and direct mapping pathway between the independent and dependent variables based on a combination of abundant data and advanced model architectures (Feng et al., 2021; Sahoo et al., 2017; Yang et al., 2020). Also, they can provide an accurate estimation of soil moisture, though the input samples are limited (Long et al., 2019). Given this, we considered using the ML models to evaluate the feasibility of flash droughts simulation over China. Our current study is to figure out the statistical interaction between RI and the anomalies of meteorological factors, which is beneficial for understanding the physical mechanism of flash droughts.

Besides, we appreciate the reviewer for providing a good idea that we can diagnose the physical-based models using machine learning technologies to improve the performance of the former models. This might be a study direction in the future flash drought field. However, considering the reason for the unclear physical mechanism of flash droughts, we applied the ML models to analyze the potential relationship between the RI and anomalies of climate factors in the current work. This is the first step before we effectively construct physical-based models to simulate and predict the RI in the future. The advantages for using machine learning algorithms have been supplemented in the introduction section in Page 4 Lines 105-111:

"*These ML technologies have superiorities in providing a fast and direct mapping pathway between the independent and dependent variables without further a priori knowledge about, or assumptions on, underlying physical processes (Feng et al., 2021; Sahoo et al., 2017; Yang et al., 2020). They can capture key information hidden in historical data, and then apply these patterns to predict target data in future scenarios.*

*Also, they can provide an accurate estimation of soil moisture, though the input samples are limited (Long et al., 2019; Almendra-Martín, et al., 2021). However, limited studies focused on flash droughts simulation based on ML technologies.*"


*"In this method, we extracted flash droughts from the entire period of records, the*

*main reasons are listed: Firstly, our method relies on continuous time series of soil moisture percentile. The intermittent data makes it hard to capture the onset, or termination of drought events accurately, and the continuity and integrity of the datasets are important for identifying the development process of drought. Secondly, enough important information related to flash droughts need be included in the ML models because flash droughts may coexist with the seasonal drought and cross-seasonal drought due to the diverse climatic conditions and underlying surface (i.e., the soil texture and vegetation cover) of China (Liu et al., 2020b). Thirdly, the occurrence of flash drought in winter is limited, which may have tiny influences on the simulation results."*

[Figure]

Fig.2 Spatial distribution of frequency of occurrence of flash droughts in different seasons over China


**Response:**

We thank the reviewer for their work and the positive comments. Yes. Flash droughts and slowly-evolving droughts are relatively homogenous and separable with respect to these predictors. Figure 3 shows the anomalies of meteorological elements (i.e., average temperature (Tmean), maximum temperature (Tmax), potential evapotranspiration (PET), precipitation (P), and relative humidity (RHU)) at the onset phase of flash droughts and traditional droughts across China. It shows that the climate driving of two types of droughts is significantly different. For energy-related meteorological elements, their average anomalies of flash droughts are more than 1-fold of standard deviation, which is generally larger than that of conventional droughts. As for the moisture-related climate factors, their anomalies of rapid intensification droughts are lower than that of slowing developing droughts. For the RI threshold method used in this study, we first identified drought events and calculated the decline rate of soil moisture. Our identification method followed the suggestion of Otkin et al. (2018) was similar to the previous literature (Ford et al., 2015; Yuan et al., 2017) which focused on two key characteristics of flash drought,

namely the intensification rate to reflect how fast the drying status proceeds, and the upper (40th) and lower (20th) limits of soil percentile to guarantee the event really falls into drought. Then, the RI of different drought events (including both flash droughts and traditional slowly-evolving droughts, and flash droughts can be distinguished from conventional droughts based on the RI threshold of "-6.5th percentile/week"), as well as relevant predictors, were employed as inputs to the ML models. Finally, the feasibilities of flash drought and slowly-evolving drought simulation were evaluated.

[Figure]

Figure 3 Meteorological anomalies of flash droughts and slowly developing droughts at the onset phase over China.

Traditional drought is influenced by a variety of predictors actively involved in the physical processes of the atmosphere, ocean, and land (Hao et al., 2018), which bring

great challenges for the prediction of drought. These predictors can be divided into three types: (1) The first type of predictors is the large-scale climate indices, for instance, Surface Sea Temperature (SST), Southern Oscillation Index (SOI), Pacific Decadal Oscillation (PDO), and North Atlantic Oscillation (NAO). The large-scale teleconnection factors have been shown to be an important driving force for the occurrence and development of drought in different areas of the world (Hoerling et al., 2003; Nicolai-Shaw et al., 2016; Trambauer et al., 2013). (2) The second type of predictor refer to the local climate variables (e.g., precipitation, temperature). For example, under the joint effects of precipitation deficit and high temperature, soil moisture may be declined and persistent moisture deficits may lead to agricultural drought (Otkin et al., 2018; Yuan et al., 2019). (3) The land initial conditions (e.g., the persistence of soil moisture) can also be used as predictors for the prediction of drought (Wu et al., 2021). Especially for flash droughts, relevant studies showed that they have a stronger meteorological driving demand than conventional droughts (Ford and Labosier, 2017; Liu et al., 2021). This suggests a close interaction between RI and these local meteorological conditions, and this may be one reason for the relatively high efficiencies of these meteorological variables for RI prediction. By contrast, the formation of traditional drought involves complicated atmosphere-land surface feedbacks at multiple scales, and it is difficult to efficiently capture the variation of RI for slowly-evolving drought from a meteorological perspective. The revisions are listed as below:

In Page 5 Line 155-158:

*"Following the suggestion of Otkin et al. (2018) and the methodology of Liu et al., (2020a), we adopt a quantitative method to identify flash droughts by focusing on the rate of intensification (RI) during their onset-development phase. The soil moisture decline rate-based approach was similar to methods of the previous literature (Ford et al., 2017; Yuan, et al., 2017)."*

In Page 26 Line 470-473:

*"For instance, precipitation deficits, enhanced evaporative demand (high temperature or heatwave), their joint or alternant effects are all possible to impose cumulative effects on soil moisture and lead to agricultural drought (Otkin et al., 2018; Yuan et al., 2019)."*

In Page 26 Line 476-484:

*"The large-scale circulation can modify precipitation's frequency and intensity,*

*increase wind speed, temperature, and evaporative demand. Several studies showed that the occurrence of droughts is related to large-scale circulation factors (Hoerling et al., 2014; Mo and Lettenmaier, 2016). Wang et al., (2016) found that under the background of El Niño of 2015/2016, a positive summer Eurasian teleconnection pattern is beneficial to anomalous northerly currents and weakening the East Asia summer monsoon, then leading to extreme droughts over northern China. The 2017 drought in north-eastern China was caused by a strong positive phase of Arctic Oscillation (AO) in March (Zeng et al., 2019). Also, 2000-2012 interdecadal drought in Eastern Africa is closely linked to the anomalies of Surface Sea Temperature (SST) in the tropical Pacific basin (Lyon and De Witt, 2012).*"

In Page 26 Line 492-494:

"*Meanwhile, they have a stronger meteorological forcing than conventional droughts (Ford and Labosier, 2017), indicating a close interaction between RI of flash drought and these local meteorological conditions. This may be one possible reason for the higher accuracies of RI prediction for flash droughts.*"

---

## Author Response (AR2)

**Response to Editor and Reviewers**

**Manuscript ID HESS-2021-541**

**Journal:** *Hydrology and Earth System Sciences*

**Article title:** Analysis of flash droughts in China using machine learning

**Author list:** *Linqi Zhang, Yi Liu, Liliang Ren, Adriaan J. Teuling, Ye Zhu, Linyong Wei, Linyan Zhang, Shanhu Jiang, Xiaoli Yang, Xiuqin Fang, Hang Yin*

Dear reviewers and editor:

Thank you so much for valuable comments and kind suggestions on our paper. Your illuminating comments and suggestions give us the possibility to properly fix several questionable issues, and to improve the overall quality of the paper. We highly appreciate your time and effort. Please find our point-to-point responses to your comments below.

**Response to reviewer #1**

The authors have addressed my questions on the first round of the manuscript. I recommend the paper for final publication in HESS.

**Response:**

We thank the reviewer for the positive comments on our study. Hope this manuscript can be accepted by HESS journal.

**Response to reviewer #2**

Zhang et al. (2022) quantified the relationship between the rate of intensification (RI) of flash drought and nine climate variables using three machine learning methods across China. This manuscript is written clearly, and it is an interesting study, particularly by linking different climate variables to the rate of intensification of drought. The results show that the random forest is preferable for estimating the flash drought rate of intensification and monitoring flash droughts in adjacent weeks of drought onset. This is my first time reviewing this manuscript. As I read the earlier discussions with Reviewers and the Author's replies, the manuscript has improved significantly since the original submission (e.g., analysis of the spatial distribution of frequency of occurrence of flash droughts in different seasons over China). The

manuscript is written clearly. I have just a few extra comments, which authors should clarify.

**Response:**

We thank the reviewer for the positive comments on this work and please see our responses in detail below.

**Comment 1:** The reanalysis ERA5-Interim soil moisture product is independent of the meteorological data used in the study. Did you consider checking the differences in the observed-based meteorological forcing data and the ERA5-Interim in-built meteorological forcing data? Additionally, please clarify why you are using ERA-Interim and why you did not use the ERA5; on the website of your data link, it is written: "ERA Interim is being phased out. Users are strongly advised to migrate to ERA5."

**Response:**

Thanks for your comment. Indeed, we used observed meteorological forcing in this study, though the ERA-Interim product has its meteorological forcing data. There are two reasons: On the one hand, ERA-Interim in-built meteorological forcing data in China were constructed based on a few meteorological stations, while 756 national observations were employed in this study. On the other hand, we considered moisture-limited and energy-limited factors to reflect meteorological driving forces conditions, while some of these data (e.g., potential evapotranspiration, relative humidity) were not included in the ERA-Interim dataset. Based on the above consideration, we used ground-based meteorological forcing data.

About the reason we chose ERA-Interim soil moisture in this study, we have three points to explain. Firstly, ERA-Interim and ERA5 soil moisture show good consistency in temporal against in-situ soil moisture. In the study of Ling et al., (2021), they evaluated the difference between satellite remote sensing (i.e., ESA CCI) and global reanalysis of soil moisture datasets (i.e., ERA-Interim, and ERA5). They found that both ERA-Interim and ERA5 soil moisture can reflect the tendency of time series and display a good agreement with observed stations relative to ESA CCI soil moisture. Secondly, we converted the original soil moisture to soil moisture percentile in this work, which alleviated the difference between ERA-Interim and ERA5 soil moisture values. Flash drought events were identified based on soil moisture percentile. Finally, this manuscript mainly focuses on evaluating the performance of

machine learning (i.e., MLR, LSTM, and RF) on flash drought simulation from the perspective of meteorological driving forces. Therefore, the regulation and conclusion are not significantly changed even if we selected ERA-Interim soil moisture to conduct this study. We carefully considered your suggestions and will replace the ERA-Interim with ERA5 soil moisture in future research. Thanks again for your valuable comments.

Page 5 Line 143-145:

"*Meanwhile, ERA-Interim SM data were converted into SM percentile to identify flash droughts over China, which alleviates the influence of soil moisture value on identification results.*"

Reference:

Ling, X., Huang, Y., Guo, W., Wang, Y., Chen, C., Qiu, B., Ge, J., Qin, K., Xue, Y., Peng, J.: Comprehensive evaluation of satellite-based and reanalysis soil moisture products using in situ observations over China, Hydrol. Earth Syst. Sci., 25, 4209–4229, https://doi.org/10.5194/hess-25-4209-2021, 2021.

**Comment 2:** I missed some discussion; if you tried to link the results of your study with the impacts and mention in the discussion/outlook, how the results of your study can be linked with impacts on agricultural production. Is the flash drought more impactful than the slowly evolving drought?

**Response:**

Thanks for your comments. As we know, flash drought is a rapid onset and high-intensity extreme drought. Its onset and development generally require the precipitation deficit along with other climate anomalies (e.g., high temperature, strong wind, and enough sunshine) that enhanced evaporative demand (Otkin et al., 2013; Anderson et al. 2013). These meteorological factors work together to quickly decrease soil moisture, gradually increase vegetation stress, and further cause the onset of flash drought (Hunt et al., 2009; Ford et al., 2015). This situation easily occurs during the growing season of vegetation and crops with the highest evaporative demand. If flash drought occurred during the critical phase of crop development, such as pollination in corn and the grain filling stage in soybeans, it would lead to large losses in agricultural production (Otkin et al., 2013; Hunt et al., 2014). For example, the 2012 flash drought in the Midwest of the U.S. aroused much attention, because this

expensive natural disaster caused about 7.62 billion dollars in agricultural losses (Hoerling et al., 2014). Thus, the occurrence of flash drought poses a potential threat to agricultural production.

There is a discrepancy between the effects of flash drought and slowly evolving drought on agricultural production. With the rapid onset of flash droughts, farmers and ranchers had little time to prepare for its detrimental effects, thus, it may result in a large reduction in crop yield (Otkin et al., 2016). While long-last traditional droughts had a persistent adverse impact on agricultural production. Therefore, it is hard to say that flash drought is more impactful that slowly developing drought on agricultural production. We need to conduct a comprehensive evaluation according to the actual drought situation. However, both flash drought and conventional drought have negative effects on agriculture, therefore, it is of significance to monitor and simulate them. In the revised manuscript, we added a new paragraph to discuss the effects of flash drought on agricultural production (Page 29-30 Lines 543-559):

*"5.4 Impact of flash droughts on agricultural production*

*Flash drought is a rapid onset and high-intensity extreme drought. Its onset and development are generally not only due to the precipitation deficit but also owe to other meteorological anomalies (e.g., high temperature, strong wind, and abundant sunshine) that enhanced evaporative demand (Otkin et al., 2013; Anderson et al. 2013). These moisture-limited and energy-limited factors work together to quickly decrease soil moisture, gradually increase vegetation stress, and then induce the onset of flash drought (Hunt et al., 2009; Ford et al., 2015). This situation is most likely to occur during the growing season of vegetation and crops with the highest evaporative demand. When flash drought occurs during the critical stage of crop development (e.g., pollination in corn and the grain filling stage in soybeans), it may lead to a large agricultural reduction (Otkin et al., 2013; Hunt et al., 2014). For instance, the 2012 flash drought in the Midwest of the U.S. was an expensive natural disaster with agricultural losses of about 7.62 billion dollars (Hoerling et al., 2014). All in all, the occurrence of flash drought poses a potential threat to agricultural production. It is worth mentioning the effects of flash drought on crop yield are different from that of conventional drought. With the rapid onset of flash drought, farmers and ranchers had limited time to prepare for its detrimental effects, thus, it may result in a large reduction in crop yield (Otkin et al., 2016). While long-last traditional drought had persistent adverse impacts on agricultural production. Generally, the impact of flash*

*drought on agricultural production is more severe than slowly developing droughts during a short period. However, it is necessary to conduct comprehensive evaluations on their effects combined with the actual drought status and background field. In addition, the accurate prediction of the RI of these droughts will be contributed to mitigating the negative impact of flash droughts on agriculture."*

References:

Anderson, M. C., Hain, C., Otkin, J., Zhan, X., Mo, K., Svoboda, M., Wardlow, B., Pimstein, A.: An intercomparison of drought indicators based on thermal remote sensing and NLDAS-2 simulations with US Drought Monitor classifications, J. Hydrometeorol., 14(4), 1035-1056, https://doi.org/10.1175/JHM-D-12-0140.1, 2013.

Ford, T. W., McRoberts, D. B., Quiring, S. M., Hall, R. E.: On the utility of in situ soil moisture observations for flash drought early warning in Oklahoma, USA, Geophys. Res. Lett., 42:9790–9798, https://doi.org/10.1002/2015GL066600, 2015.

Hoerling, M., Eischeid, J., Kumar, A., Leung, R., Mariotti, A., Mo, K., Schubert, S., Seager, R.: Causes and Predictability of the 2012 Great Plains Drought, Bull. Amer. Meteor. Soc., 95, 269–282, https://doi.org/10.1175/BAMS-D-13-00055.1, 2014.

Hunt, E. D., Hubbard, K. G., Wilhite, D. A., Arkebauer, T. J., Dutcher, A. L.: The development and evaluation of a soil moisture index, Int. J. Climatol., 29(5), 747–759, https://doi.org/10.1002/joc.1749, 2009.

Hunt, E. D., Svoboda, M., Wardlow, B., Hubbard, K., Hayes, M., Arkebauer, T.: Monitoring the effects of rapid onset of drought on non-irrigated maize with agronomic data and climate-based drought indices, Agric. For. Meteor., 191, 1–11, https://doi.org/10.1016/j.agrformet.2014.02.001, 2014.

Otkin, J. A., Anderson, M. C., Hain, C., Svoboda, M.: Examining the Relationship between Drought Development and Rapid Changes in the Evaporative Stress Index, J. Hydrometeor., 15, 938–956, https://doi.org/10.1175/JHM-D-13-0110.1, 2013.

Otkin, J. A., Anderson, M. C., Hain, C., Svoboda, M., Johnson, D., Mueller, R., Tadesse, T., Wardlow, B., Brown, J.: Assessing the evolution of soil moisture and vegetation conditions during the 2012United States flash drought, Agric. For. Meteor., 218–219, 230–242, http://dx.doi.org/10.1016/j.agrformet.2015.12.065, 2016.

**Comment 3:** Last but not least: Code and data availability statement is missing in your manuscript. Please make sure that for reproducibility, you make your analysis available to general public.

**Response:**

Thanks for pointing out this. In the revised manuscript, we have added the data availability statement.

Page 30 Line 589-593:

*"Data availability statement*

*ERA-Interim SM data used in this study are available through https://apps.ecmwf.int/datasets/data/interim-full-daily/levtype = sfc/. ERA-Interim SM data are gradually being superseded by the ERA5 reanalysis (https://cds.climate.copernicus.eu/cdsapp#!/dataset/reanalysis-era5-land).*

*Meteorological observation records can be downloaded from the China Meteorological Administration website (CMA, http://data.cma.cn/)."*

**Others:** Line 61: typo: researches => researchers

Line 188: blow => below

Line 188: decreases => decrease

Figure 1: correct typo in figure legend: "anmoly" => "anomaly"

Line 195: represent => represents

Line 277: not => do not

Line 283: captured => are captured

Line 294: were serves => served

Line 347: remove "model"

Line 362: were => was

**Response:**

Thanks for pointing out this. All these mistakes have been corrected, for example, the typo in Fig. 1 legend has been revised as below. In addition, we have checked other mistakes throughout the original manuscript and recorded them in the revised manuscript.

[Figure]

Fig.1 A concept map for identifying flash droughts.